# HOW FAR ARE TODAY'S TIME-SERIES MODELS FROM REAL-WORLD WEATHER FORECASTING APPLICATIONS?

## ABSTRACT

The development of Time-Series Forecasting (TSF) techniques is often hindered by the lack of comprehensive datasets. This is particularly problematic for time-series weather forecasting, where commonly used datasets suffer from significant limitations such as small size, limited temporal coverage, and sparse spatial distribution. These constraints severely impede the optimization and evaluation of TSF models, resulting in benchmarks that are not representative of real-world applications, such as operational weather forecasting. In this work, we introduce the WEATHER-5K dataset, a comprehensive collection of observational weather data that better reflects real-world scenarios. As a result, it enables a better training of models and a more accurate assessment of the real-world forecasting capabilities of TSF models, pushing them closer to in-situ applications. Through extensive benchmarking against operational Numerical Weather Prediction (NWP) models, we provide researchers with a clear assessment of the gap between academic TSF models and real-world weather forecasting applications. This highlights the significant performance disparity between TSF and NWP models by analyzing performance across detailed weather variables, extreme weather event prediction, and model complexity comparison. Finally, we summarise the result into recommendations to the users and highlight potential areas required to facilitate further TSF research. The dataset and benchmark implementation will be publicly available.

## 1 INTRODUCTION

Global Station Weather Forecasting (GSWF) is essential for providing precise and timely weather information, with significant implications across various sectors such as aviation Gultepe et al. (2019), agriculture Ukhurebor et al. (2022), and energy Dehalwar et al. (2016). In particular, reliable forecasts are crucial for early warning systems, aiding in the preparation for natural disasters and extreme weather events, thereby safeguarding lives and property Wang et al. (2024b); Sillmann et al. (2017). At present, the most mature and mainstream approach to precise station-based weather forecasting is the use of Numerical Weather Prediction (NWP) models. These models can be further divided into physically-based NWP models Phillips (1956); Lynch (2008) and data-driven NWP models Lam et al. (2023); Bi et al. (2023); Chen et al. (2023); Han et al. (2024a). Despite their widespread deployment and reputation as the most accurate algorithms for station weather forecasting, NWP models are inherently computationally intensive. Specifically, the execution of these algorithms depends on vast resources for data assimilation and medium-range forecasting.

On the other hand, Time-Series Forecasting (TSF) methods, which offer a more economical solution, have also attracted attention from the weather forecasting community. Recently, many TSF methods have been developed and have demonstrated significant performance on small-scale weather station datasets. For instance, methods such as those presented in (Wu et al., 2021; Zhou et al., 2021; 2022; Zeng et al., 2023; Liu et al., 2024b) have achieved exceptionally low normalized errors (e.g., below 0.23 for temperature predictions). Despite their substantial success in research-oriented domains, the applicability of these methods for real-world weather forecasting remains under-explored. Specifically, most of these methods are trained and tested on single-station datasets Wu et al. (2021) or localized regions Godahewa et al. (2021), thereby limiting their generalizability to other stations. As depicted in the right panel of Figure 1, which tracks a winter storm forecast three days in advance, the TSF method exhibits a significant performance gap when tested on unseen stations compared to the real-world weather forecasts generated by NWP model.

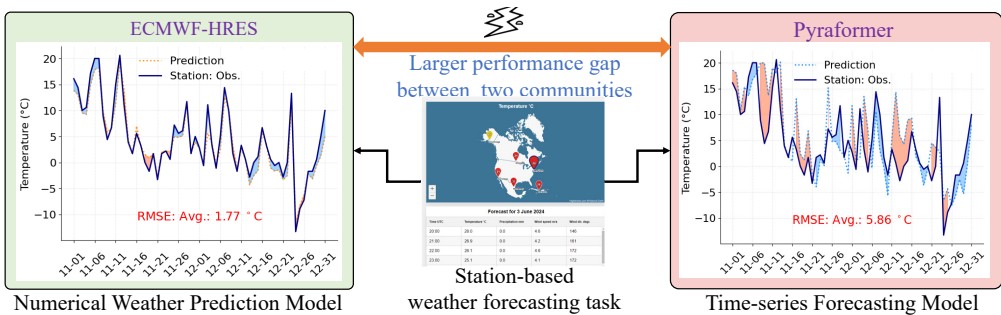

Figure 1: A case study of winter storm forecasting using methods from two communities: NWP model and TSF model. It demonstrates a big performance discrepancy between them.

Hence, an emergent question arises: *Are today's time-series models ready for deployment in real-time weather forecasting?* If not, how far are they from real-world applicability, and what can be done to improve them? To answer it, this work will deeply investigate three key aspects:

**1) Developing a Large-Scale, Comprehensive Global Station Weather Dataset.** To fully unlock the potential of existing TSF methods and substantially improve their generalization ability. Specifically, we propose WEATHER-5K, a large-scale time-series dataset for sparse weather forecasting, comprising 5,672 weather stations worldwide. This dataset provides diverse weather conditions and 10 years of hourly data per station, enabling long-term pattern analysis and robust forecasting model development. Also, we conduct thorough data analysis to uncover trends, patterns, and correlations.

**2) Conducting Extensive Assessments Between Existing TSF Algorithms and Real-World Applications Using NWP as Reference Models**. This assessment will extend beyond those shallow attempts such as Wu et al. (2023), which only compared TSF methods to NWP models within a one-day lead time and did not use the most accurate real-time forecasting products (e.g., ERA5 prediction). Here, we implement a comprehensive set of widely recognized time-series forecasting methods across domains like traffic, weather, and electricity prediction. To this end, we establish a standardized evaluation framework to conduct extensive benchmark experiments on WEATHER-5K,

**3) Fostering the Development of TSF for Real-World Weather Forecasting by Identifying future opportunities.** We have at least four recommendations for future research:

- The need to prioritize improving the long-term forecasting performance of TSF methods for their practical application in general weather forecasting.
- Besides general weather forecasting, we find that existing TSF models still lag significantly in predicting extreme weather events, which require more focused attention.
- Large-parameter models do not necessarily perform better. In our benchmarks. So developing efficient time-series modeling algorithms may yield greater benefits than merely increasing model complexity.
- Leveraging TSF methods to integrate NWP forecasts as prior knowledge can effectively improve long-term forecasting capabilities.

Overall, we believe that our dataset and benchmarks will greatly facilitate future TSF research in weather forecasting. Our efforts will help researchers better evaluate and compare different algorithms, ultimately leading to improved forecasting techniques.

## 2 RELATED WORK

**Time-series forecasting.** Time-series forecasting involves many domains Godahewa et al. (2021); Chen et al. (2017); Bai et al. (2020); Qiu et al. (2024) and their methods have undergone three stages since its birth: statistical learning, machine learning, and deep learning.

Statistical learning methods, including ARIMA Box & Pierce (1970), ETS Hyndman et al. (2008), StatsForecast Federico Garza (2022), VAR Godahewa et al. (2021), and Kalman Filter (KF) Harvey (1990), are among the early proposals and are widely utilized. These methods rely on historical data to predict future values and are based on the assumption that past observations hold predictive power.

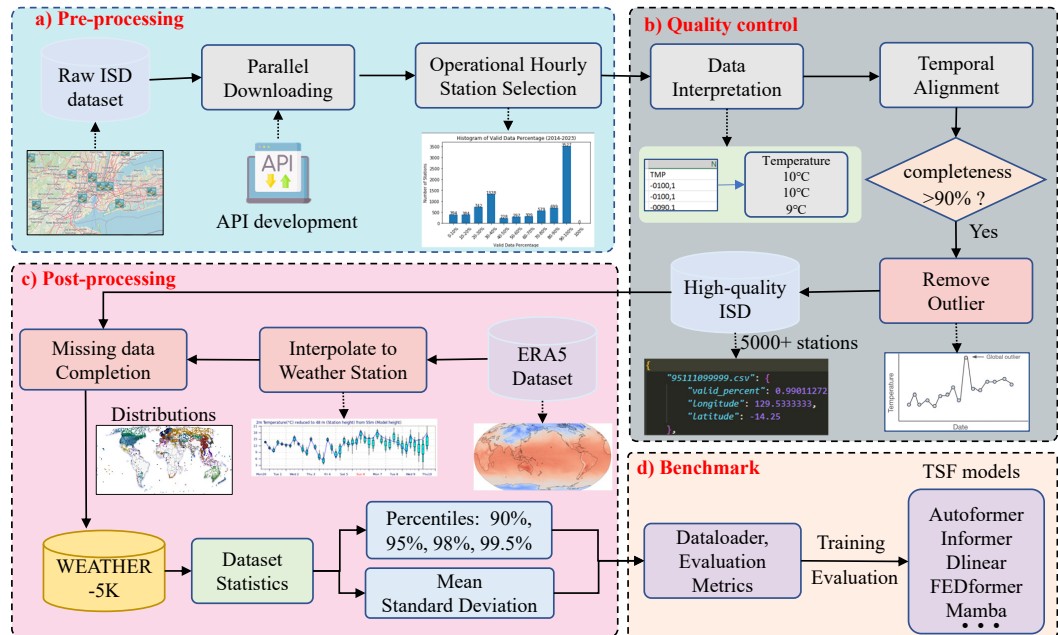

Figure 2: Flow diagram of the benchmark. a) Developing a downloading API to retrieve the raw ISD data and then do some pre-posting processing. b) Conducting rigorous quality control on selected stations to obtain a high-quality ISD subset. c) Using ERA5 to complete some missing data in the selected stations, which ensures 100% data completeness for training TSF models. d) Training and evaluating some main-stream TSF models with the basic metrics and a new proposed SEDI metric for extreme value evaluation.

While they provide a solid foundation, their performance may be limited when faced with complex patterns or nonlinear relationships.

Machine learning methods have gained prominence in time-series forecasting due to rapid advancements in the field. Algorithms such as XGBoost Chen & Guestrin (2016), GBRT Friedman (2001), Random Forests Breiman (2001), and LightGBM Ke et al. (2017) offer enhanced capabilities to handle nonlinear relationships and complex patterns. These methods demonstrate flexibility in handling different types and lengths of time-series data and generally provide superior forecasting accuracy compared to traditional statistical methods.

Leveraging the representation learning capabilities of deep neural networks, Deep-Learning (DL) methods have shown promising results in time-series forecasting. DL treats time-series as sequences of vectors and utilizes architectures such as convolutional neural networks (CNNs) Lim & Zohren (2021), recurrent neural networks (RNNs) Hewamalage et al. (2021), or Transformers Wen et al. (2022) to capture temporal dependencies. For example, TCN Bai et al. (2018) and DeepAR Salinas et al. (2020) implement CNNs or RNNs to model the temporal structure of the data. Transformer architectures, like REformer Kitaev et al. (2020), Informer Zhou et al. (2021), Pyformer Liu et al. (2021), FEDformer Zhou et al. (2022), Autoformer Wu et al. (2021), Triformer Cirstea et al. (2022), and PatchTST Nie et al. (2022), have also been applied in time-series forecasting tasks, allowing for the capture of more complex temporal dynamics and significantly improving forecasting performance. In addition, while pursuing forecasting accuracy with complex models, MLP-based models such as N-HiTS Challu et al. (2023), N-BEATS Oreshkin et al. (2019), and DLinear Zeng et al. (2023) employ a straightforward architecture with a relatively low number of parameters while achieving competitive performance. Recently, Mamba Gu & Dao (2023), a selective state space model, has also gained traction due to its ability to process dependencies in sequences while maintaining near-linear complexity. Some variants of Mamba Wang et al. (2024c); Ahamed & Cheng (2024); Shi (2024) have been successfully applied to time-series forecasting.

**Data-driven numerical weather prediction.** Since 2022, there has been a growing interest in data-driven Numerical Weather Prediction (NWP) models within the AI and atmospheric science

communities. The correlations between NWP and GSWF are as follows: 1) GSWF can be obtained by interpolating the forecast results of NWP models to specific latitudes and longitudes. Similarly, GSWF can also be used to bias-correct the forecast results of NWP models. These models, like, Pangu-Weather Bi et al. (2023), GraphCast Lam et al. (2023), FengWu Chen et al. (2023), and FengWu-GHR Han et al. (2024a), have shown the potential to outperform traditional physical-based NWP models in terms of forecast skill and operational efficiency. However, these models, operating at the mesh space ( *e.g.*, the grid resolution of $0.25°$ and $0.09°$), are may not be the optimal solution for GSWF as discussed in Section 1. Prior to our work, some initial attempts, like Corrformer Wu et al. (2023), have treated GSWF as an independent forecasting task, demonstrating promising results but remaining a great gap compared with the data-driven NWP models.

## 3 WEATHER-5K: GLOBAL STATION WEATHER DATASET

### 3.1 COLLECTION AND PRE-PROCESSING

**Data source.** WEATHER-5K is derived from global near-surface in-situ observations, which is achieved in the publicly available Integrated Surface Database (ISD), leveraging data from high-quality observation networks. ISD is a global repository of hourly and synoptic surface observations, which encompasses a wide range of meteorological parameters. In this study, we study the common variables: the wind speed and direction, temperature, dew point, and sea level pressure.

**Station selection.** ISD contains records from over $20,000$ stations spanning several decades, though certain stations are no longer operational, many do not report data on an hourly basis, and numerous stations have missing values for critical weather elements. To enhance data quality, a necessary meticulous selection process was conducted to include only long-term, hourly reporting stations that are currently operational and provide essential observations. As a result, there are $10,701$ operational stations from 2014 to 2024.

### 3.2 QUALITY CONTROL

A high-quality dataset is crucial for TSF. As shown in Figure 2, WEATHER-5K has been subjected to rigorous post-processing and quality control to ensure the reliability of the dataset.

**Data interpretation.** Due to the fact that each variable in the original ISD data is not directly recorded as floating-point data, but rather as strings, where each string encapsulates the numerical value, quality flag, reporting type, etc. of the variable. For example, a temperature field $< +0130, 1 >$ means the temperature value is 13.0 C$^\circ$, and it is at the first quality level. Therefore, the first step is to parse each variable according to the provided guidelines into a data format that is convenient for our understanding and storage.

**Temporal alignment.** The aforementioned pre-processing has selected stations that report hourly data. However, some stations may not report at 1:00, 2:00, ..., and are with a little time shift from the hour, such as 0:54, and 2:05. To address this, we merge the latest time as the hour to keep the temporal consistency. In implementation, we develop an algorithm that estimates missing hourly data points using the nearest available time points within a 30-minute window, significantly improving the distribution of valid hourly data. Despite this improvement, a tiny portion of hourly data points remained missing due to the lack of observations within the 30-minute window. To fill these gaps, linear interpolation was employed using data from the 12 consecutive hours surrounding each missing point, ensuring the reliability of the interpolated data.

**Completeness filtering.** To obtain a high-quality dataset of weather stations, only stations with more than 90% valid hourly data were chosen as the final candidates. As a result, $5,672$ weather stations worldwide are selected, spanning the period from 2014 to 2023, ensuring a recent and relevant time frame. This selection process focused on balancing the longevity of station operation, hourly data availability, and the inclusion of diverse weather variables.

**Outlier detection.** The outlier detection process begins by meticulously examining the temporal dynamics of the dataset, focusing on identifying data points that lie far outside the expected range or

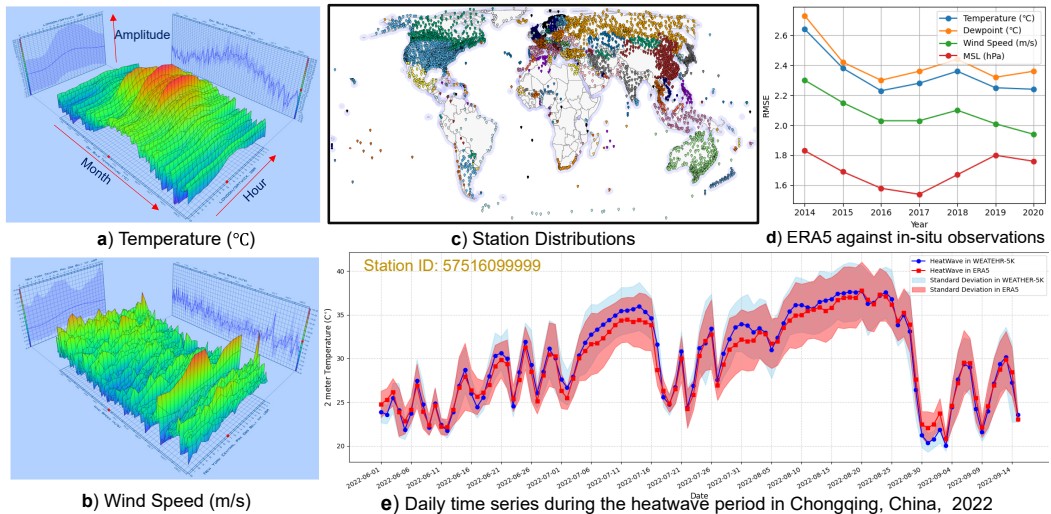

Figure 3: a) and b) The visualization of time-series data over a year. c) The geographical distribution of the weather stations in WEATHER-5K. d) The error between the observations and the ERA5 dataset. e) The daily 2m temperature at station 57516099999 in Chongqing City from 1st June to 15th September, where filled areas represent the variance from the daily mean.

exhibit unusual behaviors compared to the majority of observations. We use statistical techniques and machine learning algorithms to differentiate genuine anomalies from noise or data errors.

As the quality control process, only $5,672$ weather stations worldwide are selected, spanning the period from 2014 to 2023, ensuring a recent and relevant time frame. This selection process focused on balancing the longevity of station operation, hourly data availability, and the inclusion of diverse weather variables.

### 3.3 POST-PROCESSING

There are still very few missing data points after quality control. This is mainly due to some stations experiencing long-term missing data (such as exceeding a full day). However, ensuring the completeness of data for all stations is particularly crucial for spatial modeling with the TSF algorithms. Therefore, we interpolate the ERA5 dataset Hersbach et al. (2018) to fill in the missing data points at the station locations. ERA5 is the most widely used high-quality reanalysis dataset globally, thus introducing only minor errors. However, since wind is a high-frequency parameter, the impact of using ERA5 for wind speed interpolation will be most significant.

To facilitate training on WEATHER-5K dataset, we further calculate the mean and variance of each variable over a decade to standardize the input data. Additionally, we observed a lack of evaluation metrics for extreme values in current TSF assessment methods, especially considering the crucial role of extreme values in weather forecasting. Therefore, we also computed percentiles at different thresholds ($90\%, 95\%, 98\%, 99.5\%$) for each variable to propose metrics for extreme values.

### 3.4 QUALITATIVE ANALYSIS

**Distribution disparities of stations.** The WEATHER-5K dataset reveals regional disparities in the distribution of weather stations, which can significantly impact the learning and understanding of atmospheric dynamics in certain areas. As illustrated in Figure 3 **c)**, some land regions have sparse data coverage compared to others. These disparities can be attributed to factors such as geographical characteristics, levels of economic development, and the strategic placement of weather stations. Note that the number of oceanic stations is also very limited due to the expensive cost of establishing stations at sea. Regions with limited station coverage may exhibit unique weather patterns and phenomena that are not adequately captured due to insufficient data. Addressing these disparities and expanding coverage in underrepresented areas is crucial for improving the accuracy and reliability of weather forecasting and analysis in those regions.

Table 1: Statistics of different time-series datasets.'N/A' means the dataset is not publicly available.

| Dataset | Domain | Frequency | Lengths | Stations | Variables | Year | Volume |
|---|---|---|---|---|---|---|---|
| Exchange | Exchange | 1 day | 7,588 | 1 | 8 | 1990-2010 | 623KB |
| Electricity | Electricity | 1 hour | 26,304 | 321 | 1 | 2016-2019 | 92MB |
| ETTm2 | Electricity | 15 mins | 57,600 | 1 | 7 | 2016-2018 | 9.3MB |
| Traffic | Traffic | 1 hour | 17,544 | 862 | 1 | 2016-2018 | 131MB |
| LargeST-CA | Traffic | 5 mins | 525,888 | 8600 | 1 | 2017-2021 | 36.8GB |
| Solar | Weather | 10 mins | 52,560 | 137 | 1 | 2006 | 8.3MB |
| Wind | Weather | 15 mins | 48,673 | 1 | 7 | 2020-2021 | 2.7MB |
| Weather | Weather | 10 mins | 52,696 | 1 | 21 | 2020 | 7.0MB |
| Weather-Australia | Weather | 1 day | 1,332∼65,981 | 3,010 | 4 | unknown | 202MB |
| GlobalTempWind | Weather | 1 hour | 17,544 | 3,850 | 2 | 2019-2020 | 1034MB |
| CMA_Wind | Weather | 1 hour | 17,520 | 34,040 | 1 | 2018-2019 | N/A |
| WEATHER-5K | Weather | 1 hour | 87,648 | 5672 | 5 | 2014-2023 | 40.0GB |

**Data patterns over one year.**    Here we present a statistical analysis for different characteristics of different variables and the discrepancy of the same variable over different latitudes based on one year of data, By visualizing the two noteworthy observations: temperature, and wind speed. There are two primary patterns among different variables: First, as shown in Figure 3 **a)**, *temperature* shows a seasonal pattern. Second, Figure 3 **b)** indicates wind speeds are non-stationary, characterized by intense fluctuations and a lack of clear patterns, making them challenging to predict.

**Compare in-situ observations with ERA5.**    ERA5 is a simulated dataset rather than being based on in-situ observations, which can limit its applicability in real-world scenarios. In contrast, WEATHER-5K dataset is derived from in-situ observations. To demonstrate their difference, we calculate the RMSE of ERA5 against in-situ observations (Jiao et al., 2021), as shown in Figure 3 **d)**, which highlights an unacceptably high RMSE when comparing ERA5 data to real-world observations.

**Heatwave analysis in China, 2022.**    Figur 3 **e)** illustrates the daily temperature variations at Chongqing city in China (57516099999) from June to September in 2022. From this event, we can see that both the average and the maximum temperatures of WEATHER-5K are greater than those of in ERA5 in most of the heatwave days. This indicates that ERA5 consistently underestimated the diurnal temperature range at this station throughout the heatwave period.

**Comparison with existing datasets.** Table 1 presents a comparison between the WEATHER-5K and other popular TSF datasets, which shows several limitations in previous datasets. 1) *Small scale and out-of-date*: The mainstream time-series datasets (Lai et al., 2018; Trindade, 2015; Wu et al., 2021) used for research purposes remain relatively small in scale. For instance, datasets related to electricity consumption or exchange rates are sparse or outdated, which limits the practical application of forecasting models. 2) *Lagging behind other fields*: The TSF community has been slower in incorporating large-scale datasets. The use of extensive datasets (*e.g.*, Common Crawl and LAION-5B Schuhmann et al. (2022)) in other fields have demonstrated unprecedented economic value and significantly advanced scientific discoveries. However, until recently, the first large-scale time-series dataset, LargeST Liu et al. (2024a), is only introduced. The proposed WEATHER-5K will address the limitation of small-scale time series weather datasets. This abundance of data enables researchers to tackle more complex forecasting challenges. See Section C for more dataset analysis.

## 4    TIME-SERIES FORECASTING BENCHMARKS ON WEATHER-5K

### 4.1    PROBLEM DEFINITION

Considering $N$ weather stations around the world and each station collects $V$ meteorological variables, the data of all weather stations can be represented by a spatial-temporal time-series $X \in \mathbb{R}^{N \times T \times V}$ for a given look-back window of fixed length $T$. At timestamp $t$, time-series forecasting is to predict $\hat{X}_{t+1:t+\tau} = \{X_{t+1}, ..., X_{t+\tau}\}$ based on the past $T$ frames $X_{t-T+1:t} = \{X_{t-T+1}, ..., X_t\}$. Here, $\tau$ is the length of the forecast horizon. Using $X$ and $\hat{X}$ to represent the observation data and the

forecasted data, respectively, the process of GSWF can be simplified by a mapping: $\hat{X} = \mathcal{M}(X)$, where $\mathcal{M}$ can be different kinds of time-series forecasting methods. For example, by setting $N = 1$ and ignoring the spatial information, many state-of-the-art time-series forecasting methods Li et al. (2022); Zhou et al. (2021); Liu et al. (2021); Wu et al. (2021); Zhou et al. (2022); Gu & Dao (2023); Wang et al. (2024a); Nie et al. (2022); Liu et al. (2024b) can be explored on this task. When $N$ is multiple scattered stations, method Wu et al. (2023) based on spatial-temporal modeling can also be applied to this task.

## 4.2 EVALUATION METRICS

**Overall performance.** Mean Absolute Error (MAE) and Mean Square Error (MSE) are used to evaluate the overall performance of the GSWF. MAE measures the predictive robustness of an algorithm but is insensitive to outliers, whereas MSE is sensitive to outliers and can amplify errors.

**Extreme performance metric.** In addition to standard metrics MAE and MSE, the precision in predicting extreme weather events, such as unusually high or low temperatures, is crucial in real-world applications. Hence, we introduce a specialized metric, the Symmetric Extremal Dependence Index (SEDI). Han et al. (2024b); Xu et al. (2024). It classifies each prediction in its station location as either extreme or normal weather based on upper quantile thresholds ($90\%, 95\%, 98\%$, and $99.5\%$) or lower quantile thresholds ($10\%, 5\%, 2\%$, and $0.5\%$). The calculation of SEDI value can be formulated as:

$$\text{SEDI}(p) = \frac{\text{sum}(\hat{X} < Q^p_{lower} \& X < Q^p_{lower}) + \text{sum}(\hat{X} > Q^p_{upper} \& X > Q^p_{upper})}{\text{sum}(X < Q^p_{lower}) + \text{sum}(X > Q^p_{upper})}. \quad (1)$$

where $\hat{X} < Q^p_{lower}$ and $X < Q^p_{lower}$ judge whether the predicted or observed data point belongs to the extreme event or not based on the threshold $Q^p_{lower}$, and vice versa for the upper percentiles. Here we access both two opposite percentiles for accessing extreme small value (*e.g.,* winter storm) and extreme large value (*e.g.,* heatwave). SEDI $\in [0, 1]$ quantifies the model's ability to correctly identify extreme weather events. Higher SEDI indicates better performance in extreme weather prediction.

## 4.3 EXPERIMENTAL PROTOCOLS

**Dataset splitting.** The WEATHER-5K dataset is divided into three subsets: training (with years 2014-2021), validation (with yea 2022) and test (with year 2023), which follows an 8:1:1 ratio. allowing the model to be trained on sufficient historical data, validated on a separate year, and tested on the most recent data for accurate evaluation.

**Baselines.** We here compare 9 baselines as mentioned in Section 2, which can be categorized as: **1) Temporal-only** methods, which includes popular transformer-based long-term forecasting models between 2021 and 2024: Informer (2021) Zhou et al. (2021), Autoformer (2021) Wu et al. (2021), Pyraformer (2021) Liu et al. (2021), FEDformer (2022) Zhou et al. (2022), PatchTST (2023) Nie et al. (2022), iTransformer (2024) Liu et al. (2024b), MLP-based models: DLinear (2023)Zeng et al. (2023) and the new trending architecture Mamba (2023) Gu & Dao (2023) as baselines. **2) Spatial-temporal** method: We also include a recent method, Corrformer (2023) Wu et al. (2023) as a baseline model that considers the dynamic characteristics of correlations among weather stations at different locations.

**Task settings**. To facilitate fair comparison between different baselines, we align the input length of all baselines. The final setting is to predict the $\tau$-step future based on 48 historical steps, where the input length is chosen based on the experimental results shown in Figure 4 **b)**, which illustrates the performance variation for four weather variables as the input sequence length increases. To trade off the training cost and performance, we ultimately set the input length as 48 to balance computation and performance. Specifically, we predict the future weather conditions for lead times of 1, 3, 5, and 7 days, corresponding to predicting the 24-step, 72-step, 120-step, and 168-step future data, respectively. Note the report results are only performed once instead of multiple times in original implementations. This does not affect the comparison as we observe the results are stable for different seeds due to the large dataset volume.

**Implementation details**. We develop and implement the baselines based on the Time-Series-Library (Tim). The training is performed for a total of 300,000 iterations, starting with a learning rate

Table 2: Benchmarks on our WEATHER-5K. The results are based on 4 different prediction lengths: 24, 72, 120, and 168, where the input length is 48. Those baselines are developed for specific research, such as "*Electricity (E)*", "*Exchange (X)*", "*Influenza (I)*", "*Traffic (T)*", "*Weather (W)*" and "*General (G)*" domains. ECMWF-HRES is the physical-based NWP model, representing the most accurate weather forecasting model. Underline is the second-best performance.

| Baselines | Lead Time | Temperature | | Dewpoint | | Wind Rate | | Wind Direc. | | Sea Level | |
|---|---|---|---|---|---|---|---|---|---|---|---|
| | | MAE | MSE | MAE | MSE | MAE | MSE | MAE | MSE | MAE | MSE |
| ECMWF-HRES 
 Best NWP Model | 24 | 1.76 | 7.39 | 1.85 | 7.94 | 1.48 | 4.53 | 63.8 | 7158.3 | **0.86** | **2.68** |
| | 72 | **1.87** | **8.01** | **1.94** | **8.48** | **1.52** | **4.76** | 72.4 | **8215.6** | **1.06** | **3.31** |
| | 120 | **1.99** | **8.79** | 2.14 | 10.87 | **1.58** | 5.11 | 75.4 | 8647.7 | **1.38** | **5.15** |
| | 168 | **2.15** | **10.06** | **2.32** | **12.56** | 1.66 | 5.59 | 78.3 | 8945.7 | **1.87** | **9.52** |
| Pyraformer 
 2021 
 *EW* 
 Best TSF Model | 24 | **1.75** | **6.92** | **1.83** | **7.88** | **1.30** | **3.58** | 61.8 | 6930.2 | 1.90 | 9.72 |
| | 72 | 2.47 | 13.03 | 2.67 | 15.39 | 1.52 | 4.97 | 72.0 | 8222.4 | 3.76 | 33.67 |
| | 120 | 2.77 | 16.04 | 3.00 | 18.95 | 1.59 | 5.37 | **75.1** | 8610.7 | 4.43 | 43.91 |
| | 168 | 2.95 | 17.95 | 3.20 | 21.06 | **1.61** | **5.56** | 76.4 | 8773.5 | 4.77 | 49.97 |
| Informer 
 2021 
 *EW* | 24 | 1.88 | 7.51 | 1.94 | 8.30 | 1.30 | 3.62 | **60.7** | **6906.9** | 2.01 | 10.56 |
| | 72 | 2.75 | 14.84 | 2.86 | 17.24 | 1.53 | 4.86 | **71.5** | 8251.4 | 4.24 | 39.24 |
| | 120 | 3.11 | 18.21 | 3.25 | 21.50 | 1.60 | 5.38 | 75.7 | **8504.5** | 5.15 | 54.31 |
| | 168 | 3.24 | 20.24 | 3.43 | 24.89 | 1.63 | 5.65 | **76.2** | **8718.4** | 5.26 | 58.42 |
| Autoformer 
 2021 
 *EXITW* | 24 | 1.93 | 8.64 | 2.06 | 9.57 | 1.42 | 3.97 | 66.5 | 7710.0 | 2.26 | 12.78 |
| | 72 | 2.72 | 15.14 | 2.97 | 18.38 | 1.54 | 5.14 | 75.4 | 9111.5 | 4.25 | 42.34 |
| | 120 | 3.21 | 20.27 | 3.34 | 23.12 | 1.58 | 5.73 | 79.2 | 9143.5 | 4.83 | 48.88 |
| | 168 | 3.43 | 21.71 | 3.56 | 22.55 | 1.64 | 5.95 | 79.8 | 9435.8 | 5.32 | 61.85 |
| FEDformer 
 2022 
 *EXITW* | 24 | 1.98 | 8.45 | 2.02 | 9.25 | 1.36 | 3.91 | 66.0 | 7384.1 | 2.13 | 11.43 |
| | 72 | 2.87 | 16.50 | 3.01 | 18.70 | 1.59 | 5.31 | 76.2 | 8824.8 | 4.15 | 37.60 |
| | 120 | 3.19 | 20.29 | 3.36 | 23.10 | 1.66 | 5.71 | 79.0 | 9143.3 | 4.81 | 48.86 |
| | 168 | 3.35 | 22.12 | 3.54 | 25.21 | 1.68 | 5.88 | 79.7 | 9189.2 | 5.01 | 53.39 |
| DLinear 
 2023 
 *EXITW* | 24 | 2.71 | 13.82 | 2.47 | 12.36 | 1.44 | 4.34 | 66.6 | 8234.5 | 3.09 | 21.34 |
| | 72 | 3.55 | 23.05 | 3.48 | 22.85 | 1.62 | 5.37 | 75.0 | 9250.8 | 4.64 | 45.83 |
| | 120 | 3.90 | 27.60 | 3.89 | 27.72 | 1.67 | 5.70 | 77.3 | 9510.6 | 5.19 | 56.22 |
| | 168 | 4.11 | 30.38 | 4.11 | 30.58 | 1.69 | 5.88 | 78.4 | 9630.0 | 5.48 | 61.73 |
| PatchTST 
 2023 
 *EXITW* | 24 | 2.05 | 9.26 | 2.16 | 10.58 | 1.40 | 4.20 | 66.2 | 7765.8 | 2.19 | 12.54 |
| | 72 | 2.82 | 16.60 | 3.06 | 19.96 | 1.60 | 5.39 | 75.2 | 9067.8 | 4.28 | 42.46 |
| | 120 | 3.15 | 20.32 | 3.43 | 24.39 | 1.66 | 5.79 | 77.8 | 9452.6 | 5.09 | 57.29 |
| | 168 | 3.33 | 22.54 | 3.63 | 26.94 | 1.69 | 6.00 | 79.0 | 9638.1 | 5.51 | 65.3 |
| Corrformer 
 2023 
 *W* | 24 | 1.99 | 8.21 | 2.09 | 9.47 | 1.38 | 3.83 | 66.7 | 7832.3 | 2.19 | 12.39 |
| | 72 | 2.74 | 15.16 | 2.99 | 18.40 | 1.56 | 4.91 | 75.6 | 9111.7 | 4.27 | 42.36 |
| | 120 | 3.06 | 18.63 | 3.34 | 22.48 | 1.61 | 5.56 | 78.0 | 9477.4 | 5.08 | 57.13 |
| | 168 | 3.09 | 18.69 | 3.36 | 22.53 | 1.63 | 5.69 | 78.9 | 9636.0 | 5.34 | 61.83 |
| Mamba 
 2023 
 *G* | 24 | 1.98 | 8.59 | 2.01 | 9.52 | 1.37 | 4.02 | 66.0 | 7709.5 | 2.21 | 12.73 |
| | 72 | 2.79 | 16.00 | 2.90 | 18.11 | 1.55 | 5.11 | 75.1 | 8863.9 | 4.29 | 41.88 |
| | 120 | 3.03 | 18.47 | 3.18 | 21.02 | 1.58 | 5.28 | 76.7 | 8931.2 | 4.93 | 52.56 |
| | 168 | 3.16 | 19.88 | 3.32 | 22.53 | 1.59 | **5.35** | 77.4 | 8958.8 | 5.21 | 57.37 |
| iTransformer 
 2024 
 *ETW* | 24 | 1.82 | 7.49 | 1.93 | 8.80 | 1.32 | 3.77 | 63.2 | 7358.8 | 1.99 | 10.84 |
| | 72 | 2.60 | 14.46 | 2.84 | 17.5 | 1.52 | 4.96 | 73.2 | 8713.3 | 4.14 | 40.65 |
| | 120 | 2.97 | 18.36 | 3.24 | 22.16 | 1.59 | 5.42 | 76.4 | 9192.2 | 4.95 | 54.67 |
| | 168 | 3.18 | 20.64 | 3.48 | 24.89 | 1.64 | 5.67 | 78.0 | 9441.1 | 5.36 | 62.31 |

of 1e-4. We employ the cosine decay strategy and gradually decay the learning rate to 0 by the end of training. The batch size for all models is set to 1,024 except for Correformer. During the validation phase, an early stopping is executed if training loss does not decrease for three consecutive times. The checkpoint with the lowest validation loss encountered prior to the early stop is saved and used for testing. Experiments are conducted on 224 Intel(R) Xeon(R) Platinum 8480CL CPUs @ 3.80 GHz, 2.0 TB RAM computing server, equipped with 8 NVIDIA H800 GPUs.

Table 3: Extreme weather events forecasting evaluation@SEDI (%) of NWP model and TSF-models. SEDI is calculated on the predicted results with 120 lengths. Note that the percentiles here include the lower bounds and upper bounds. Underline is the second-best performance.

| Baselines | Temperature | | Dewpoint | | Wind Speed | | Wind Direction | | Sea Level | |
|---|---|---|---|---|---|---|---|---|---|---|
| | 99.5th↑ | 90th↑ | 99.5th↑ | 90th↑ | 99.5th↑ | 90th↑ | 99.5th↑ | 90th↑ | 99.5th↑ | 90th↑ |
| ECMWF-HRES | **37.4** | **82.6** | **35.4** | **76.4** | **10.2** | **40.8** | **13.1** | **45.4** | **77.5** | **89.7** |
| Informer | 11.8 | 49.5 | 9.2 | 39.2 | 2.1 | 6.7 | 0.12 | 2.9 | 9.8 | 35.7 |
| Autoformer | 12.4 | 52.1 | 8.3 | 38.9 | 0.3 | 7.8 | 0.13 | 1.6 | 10.4 | 32.1 |
| Pyraformer | 10.7 | 54.8 | 7.2 | 40.1 | 0.6 | 7.2 | 0.06 | 1.1 | 10.5 | 26.2 |
| FEDformer | 11.9 | 50.9 | 9.9 | 40.7 | 2.9 | 9.5 | 0.08 | 0.7 | 7.5 | 21.4 |
| DLinear | 5.8 | 18.8 | 3.2 | 19.9 | 0.3 | 5.1 | 0.13 | 1.7 | 2.8 | 17.5 |
| PatchTST | 10.9 | 50.8 | 8.9 | 42.4 | 0.5 | 8.9 | 0.10 | 2.2 | 13.5 | 36.7 |
| Corrformer | 10.9 | 48.9 | 8.4 | 39.9 | 1.7 | 8.4 | 0.12 | 0.9 | 8.9 | 30.9 |
| Mamba | 10.0 | 51.3 | 7.5 | 40.6 | 0.9 | 8.1 | 0.05 | 1.0 | 10.1 | 31.3 |
| iTransformer | 14.1 | 55.0 | 10.4 | 44.8 | 1.3 | 10.3 | 0.14 | 2.3 | 15.9 | 37.5 |

## 4.4 OBSERVATIONS AND FINDINGS

**RQ1: How do TSF models compare with NWP models in terms of general performance?** . In long-term predictions, where lead time $\geq 72$ hours, the NWP model ECMWF-HRES (EC) is overall superior to all existing TSF methods across almost all variables apart from wind speed and direction, while in the short-term prediction, where lead time $= 24$ hours, some TSF methods (Pyraformer Liu et al. (2021)) shows a comparable performance, which shows there is a long way for TSF models to be the applicable model for weather forecasting. Details can be found in Table 2, which reports the MAE and MSE for five variables under different predictive lengths. We also found that the simple linear implementation DLinear Zeng et al. (2023) shows a poor performance while the early method, like Pyraformer Liu et al. (2021), shows a significant advantage among all baselines. Overall, the remaining baselines show similar performance. There could be several reasons for this. Firstly, the existing TSF models have relatively small parameter settings and computational capacity, which can achieve good fitting performance on small-scale datasets but may not be suitable for large-scale datasets. Additionally, these models, except Corrformer Wu et al. (2023), only consider temporal dependencies and overlook the spatial distribution differences and correlations in meteorological data. Furthermore, we observe that Corrformer, despite considering spatial relationships, does not perform as well as some simpler models like Informer Zhou et al. (2021). One possible reason for this may be that Corrformer's spatial modeling relies on pre-defined local regions. Moreover, we noticed that different models exhibit preferences for some variables. For example, Informer Zhou et al. (2021) performs better in predicting wind speed and direction than other methods. In addition, some models show preferences for long-term forecasting, such as the Mamba Gu & Dao (2023) model, which achieves the best performance in long-term predictions even though it has a general short-term performance. This suggests that the Mamba structure may be more suitable for long-term forecasting. Finally, based on the current results, the cumulative error stabilizes and increases significantly after the third day. This indicates the predictive errors are large after three days, and there is still ample room for improvement in long-term prediction performance.

**RQ2: Can TSF models perform well in real-world extreme weather prediction?** Another finding is that current TSF models struggle to predict extreme values, whereas numerical models excel at forecasting these extreme values, which is crucial for extreme weather event assessment. Table 3 presents a report on the predictive performance of various models for extreme values. Our finding is that the current time-series forecasting models studied in this paper can not effectively capture the extreme values, especially with the lower and upper quantiles at $0.05\%$ and $99.5\%$. Additionally, it is observed that the performance of wind prediction is the worst among all variables. This can be attributed to the non-stationary nature of the wind distribution, which makes it extremely challenging to predict accurately. The evaluation results also indicate a research direction for future time-series forecasting, which is to pay more attention to extreme values.

**RQ3: Are large-parameter TSF models necessary for weather forecasting?** Highly parameterized models with intensive computational requirements do not necessarily enhance the predictive

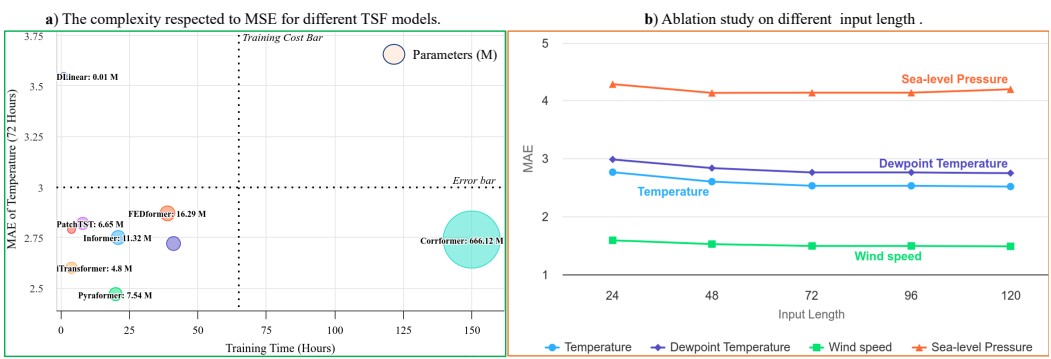

Figure 4: a) Model performance vs complexity. b) The performance impact of input length.

performance of time series forecasting. Figure 4 compares the model complexity and prediction accuracy (average results over 72 hours) of different baselines. This figure give some guidances on the future direction of TSF models. For instance, blindly scaling model parameters may not necessarily improve prediction accuracy. Pyraformer (Liu et al., 2021), for example, not only outperforms all baselines in terms of performance but also has relatively low training costs and parameter counts. On the other hand, Corrformer (Wu et al., 2023), with its high parameter count, significantly increases training time without bringing about substantial performance improvements. Since time series models are typically applied in specific domains and the data is updated rapidly, a model with fewer parameters is more conducive to practical deployment and iterative updates.

**RQ4: How to improve future weather forecasting methods leveraging the advantages of TSF and NWP models?** Bridging the TSF models with NWP models would greatly enhance the performance of GSWF. We are aware that most NWP models Bi et al. (2023); Lam et al. (2023); Han et al. (2024a) can provide robust global atmospheric forecasts. By utilizing outputs from these models, we can develop bias correction models tailored to meteorological stations. Leveraging this diverse information by bridging GSWF with numerical prediction models could potentially enhance weather prediction accuracy at these stations.

## 5 CONCLUSION AND LIMITATION

To facilitate accurate, efficient, and scalable weather forecasting for global weather stations, we introduce WEATHER-5K as a new benchmark dataset. WEATHER-5K encompasses numerous global stations, providing comprehensive, long-term meteorological data. This dataset enables state-of-the-art time-series forecasting methods to be easily adopted and yield promising results. However, we also noticed that current methods might still lag behind numerical weather prediction models, particularly for longer lead times. This means WEATHER-5K would present new challenges and opportunities, fostering advanced techniques and innovative research.

**Limitation.** Currently, we have three major limitations. 1). Spatial Quality: The current version of WEATHER-5K offers observations from a global network, but coverage remains sparse in regions like Africa and South America. Future enhancements will focus on integrating data from diverse sources, such as MADIS reports (including METER and Mesonet), to achieve denser station distribution and improve evaluation accuracy. 2). Missing Data Handling. To maintain data integrity, we utilized interpolated results from the ERA5 dataset to fill in missing observations. While this interpolation introduces some errors, minimally affecting temperature, dew point temperature, and pressure but significantly impacting wind data, we ensured that only a small fraction of missing values were filled in to minimize these errors. 3). Lack of Spatial-Temporal Methods: There is a scarcity of research utilizing spatial modeling in time-series forecasting. Consequently, the baselines we implemented do not fully leverage the potential of the WEATHER-5 dataset. We look forward to future researchers fully utilizing WEATHER-5K to develop more advanced time-series forecasting methods.

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

## A  APPENDIX

## B  DATASET DOCUMENTATION

We organize the dataset documentation based on the template of datasheets for datasets Gebru et al. (2021).

### B.1  MOTIVATION

**For what purpose was the dataset created? Was there a specific task in mind? Was there a specific gap that needed to be filled? Please provide a description.**

This dataset is created with the following motivations: 1)The current weather station dataset limits the applicability of forecasting models in real-world scenarios. Hence, it is urgent to develop a comprehensive global station weather dataset that enables forecasting models to generalize across diverse stations and regions worldwide. 2) The limited sizes of existing time-series datasets may not reflect the real performance of the forecasting models, the proposed large weather station dataset can also serve as an extensive time-series dataset to perform comprehensive time-series forecasting benchmarks for various forecasting methods. 3) The existing simple datasets fail to encompass the complex scientific problems that researchers need to discover and resolve, thereby hindering progress in the field of time-series prediction. the proposed dataset offers a diverse temporal and spatial range of time-series data, enabling a comprehensive evaluation of time-series forecasting methods and driving significant advancements in the field. 4) A large-scale weather station dataset is a crucial source of observational data for numerical weather prediction models, effectively bridging the gap between numerical models and station-based predictions. This not only improves the accuracy of numerical forecasts but also plays a vital role in verifying and evaluating the predictive performance of numerical weather prediction models.

### B.2  COMPOSITION

**What do the instances that comprise the dataset represent (e.g., documents, photos, people, countries)? Are there multiple types of instances (e.g., movies, users, and ratings; people and interactions between them; nodes and edges)? Please provide a description.**

WEATHER-5K consists of 5,672 CSV files. Each CSV file represents data from a single weather station, with hourly observations recorded from 2014 to 2023. The dataset represents a collection of weather observation data, where each instance corresponds to an hourly observation from a specific weather station, with various meteorological measurements and auxiliary information.

**How many instances are there in total (of each type, if appropriate)?**

WEATHER-5K has a total number of 5,762 stations with a 10-year time coverage and includes 8 mandatory variables and 2 auxiliary features. For each sensor. It also possesses 87,648 instances.

**Does the dataset contain all possible instances or is it a sample (not necessarily random) of instances from a larger set? If the dataset is a sample, then what is the larger set? Is the sample**

**representative of the larger set (e.g., geographic coverage)? If so, please describe how this representativeness was validated/verified.**

Our dataset is collected and further processed using data sourced from the National Centers for Environmental Information (NCEI), specifically the Integrated Surface Database (ISD), [1]. Although the ISD contains records from over 20,000 stations spanning several decades, not all stations are suitable for machine learning applications. For instance, some stations are no longer operational, many do not report data on an hourly basis, and numerous stations have missing values for critical weather elements. To get a high-quality weather station dataset, a meticulous selection process was conducted to include only long-term, hourly reporting stations that are currently operational and provide essential observations such as temperature, dew point temperature, wind, and sea level pressure. After that, we use the process procedure detailed in Section 3 to make the final WEATHER-5K, which is in the principle of applicability for time-series forecasting research.

**What data does each instance consist of? "Raw" data (e.g., unprocessed text or images) or features? In either case, please provide a description.**

The key characteristics of each instance are:

**Instance Type** Each row in the CSV file represents a single hourly weather observation from a specific weather station.

**Instance Fields** Each instance (row) has the following fields:

| Field | Description |
| --- | --- |
| DATE | The date of the observation |
| LONGITUDE | The longitude of the weather station |
| LATITUDE | The latitude of the weather station |
| TMP | The temperature observation |
| DEW | The dew point observation |
| WND_ANGLE | The wind angle observation |
| WND_RATE | The wind rate observation |
| SLP | The sea level pressure observation |
| MASK | A binary list indicate the quality of the observation |
| TIME_DIFF | An auxiliary field |

**Temporal Dimension** The dataset covers hourly weather observations from 2014-01-01T00:00:00 to 2023-12-31T00:00:00, a total of 87,648 time slots.

**Spatial Dimension** Each CSV file represents data from a single weather station, identified by its geographic coordinates (LONGITUDE and LATITUDE).

**Is there a label or target associated with each instance? If so, please provide a description.**

No, weather observation data can take itself as label in the forecasting task, and weather forecasting can be considered as a self-supervised learning task.

**Is any information missing from individual instances? If so, please provide a description, explaining why this information is missing (e.g., because it was unavailable).**

No, many efforts have been made to ensure there is no missing data in the WEATHER-5K dataset.

**Are relationships between individual instances made explicit (e.g., users' movie ratings, social network links)? If so, please describe how these relationships are made explicit.**

Yes, the weather stations in the dataset have geographical relationships, and we have used latitude, longitude, and elevation to represent their geographic locations. This information can be leveraged in subsequent work to model the spatial relationships between the instances.

**Are there recommended data splits (e.g., training, development/validation, testing)? If so, please provide a description of these splits, explaining the rationale behind them.**

Yes, we chronologically split the data into train (2013-01-01 to 2021-12-31), validation (2022-01-01 to 2022-12-31), and test (2023-01-01 to 2023-12-31) sets, with a ratio of 8:1:1.

---

[1] www.ncei.noaa.gov/products/land-based-station/integrated-surface-database

**Are there any errors, sources of noise, or redundancies in the dataset? If so, please provide a description.**

Yes, the errors and noise in the dataset arise from two main sources. Firstly, the use of meteorological automatic stations introduces a certain degree of observational error, particularly in the measurement of wind speed and direction, which are relatively difficult to measure accurately. Secondly, in our data processing efforts to ensure the completeness of the dataset, we have employed interpolation operations, which can introduce some additional error. However, the proportion of error introduced by the interpolation is relatively small.

**Is the dataset self-contained, or does it link to or otherwise rely on external resources (e.g., websites, tweets, other datasets)?**

Yes, it is self-contained.

**Does the dataset contain data that might be considered confidential (e.g., data that is protected by legal privilege or by doctor–patient confidentiality, data that includes the content of individuals' non-public communications)? If so, please provide a description.**

No, all our data are from a publicly available data source, i.e., NCEI.

**Does the dataset contain data that, if viewed directly, might be offensive, insulting, threatening, or might otherwise cause anxiety? If so, please describe why.**

No, all our data are numerical.

## B.3 COLLECTION PROCESS

**How was the data associated with each instance acquired? Was the data directly observable (e.g., raw text, movie ratings), reported by subjects (e.g., survey responses), or indirectly inferred/derived from other data (e.g., part-of-speech tags, model-based guesses for age or language)?**

We source the data from the the Integrated Surface Database (ISD) ogrinized and maintained by the National Centers for Environmental Information (NCEI). ISD is a global database that consists of hourly and synoptic surface observations compiled from numerous sources into a single common ASCII format and common data model. ISD integrates data from more than 100 original data sources.

**What mechanisms or procedures were used to collect the data (e.g., hardware apparatuses or sensors, manual human curation, software programs, software APIs)? How were these mechanisms or procedures validated?**

NCEI (formerly the National Climatic Data Center) started developing ISD in 1998 with assistance from partners in the U.S. Air Force and Navy, as well external funding from several sources. The database incorporates data from over 35,000 stations around the world, with some that include having, and includes observations data from as far back as 1901. The number of stations with data in ISD increased substantially in the 1940s and again in the early 1970s. There are currently more than 14,000 active ISD stations that are updated daily in the database. The total uncompressed data volume is around 600 gigabytes; however, it continues to grow as more data are added.

**If the dataset is a sample from a larger set, what was the sampling strategy (e.g., deterministic, probabilistic with specific sampling probabilities)?**

The sampling strategy is deterministic.

**Who was involved in the data collection process (e.g., students, crowdworkers, contractors) and how were they compensated (e.g., how much were crowdworkers paid)?**

Our code collects publicly available data, which is free. On our side, we developed a download API to efficiently retrieve the source data, which was done by our team members.

**Over what timeframe was the data collected? Does this timeframe match the creation timeframe of the data associated with the instances (e.g., recent crawl of old news articles)? If not, please describe the timeframe in which the data associated with the instances was created.**

The WEATHER-5K dataset is collected and processed in 2024. This timeframe of the source data data is matches the creation timeframe of the data.

**Were any ethical review processes conducted (e.g., by an institutional review board)?**

No, such processes are unnecessary in our case.

### B.4 PREPROCESSING/CLEANING/LABELING

**Was any preprocessing/cleaning/labeling of the data done (e.g., discretization or bucketing, tokenization, part-of-speech tagging, SIFT feature extraction, removal of instances, processing of missing values)? If so, please provide a description.**

Yes, to obtain a high-quality dataset of weather stations, a series of post-processing steps were performed on the raw weather station data collected from 2014 to 2024. Initially, $10,701$ commonly operating stations were identified. The first step involved selecting stations that reported data every hour on the hour. However, many stations did not meet this criterion. To address this, a replacement method estimated missing hourly data points using the nearest available time points within a 30-minute window, significantly improving the distribution of valid hourly data. Some following processing steps are described in Section 3.

**Was the "raw" data saved in addition to the preprocessed/cleaned/labeled data (e.g., to support unanticipated future uses)? If so, please provide a link or other access point to the "raw" data.**

The raw data are available in the NCEI. The link is: `www.ncei.noaa.gov/products/ land-based-station/integrated-surface-database`. To get the preprocessed data, you can run the 'weather_station_api.py' in our final released repository.

**Is the software that was used to preprocess/clean/label the data available? If so, please provide a link or other access point.**

No.

### B.5 USES

**Has the dataset been used for any tasks already? If so, please provide a description.**

The dataset is used in this paper for the global station weather forecasting task.

**Is there a repository that links to any or all papers or systems that use the dataset? If so, please provide a link or other access point.**

No, but we may include a leader board and list papers using this dataset in the future.

**What (other) tasks could the dataset be used for?**

Weather data imputation, numerical weather prediction, and data assimilation

**Is there anything about the composition of the dataset or the way it was collected and preprocessed/cleaned/labeled that might impact future uses?**

We believe that our dataset will not encounter usage limit.

**Are there tasks for which the dataset should not be used? If so, please provide a description.**

No, users could use our dataset in any task as long as it does not violate laws.

### B.6 DISTRIBUTION

**Will the dataset be distributed to third parties outside of the entity (e.g., company, institution, organization) on behalf of which the dataset was created? If so, please provide a description.**

No, it will always be held on GitHub.

**How will the dataset will be distributed (e.g., tarball on website, API, GitHub)? Does the dataset have a digital object identifier (DOI)?**

The instructions for building WEATHER-5K will be available in the released code. The dataset does not have a digital object identifier currently.

**When will the dataset be distributed?**

On June 07, 2024.

**Will the dataset be distributed under a copyright or other intellectual property (IP) license, and/or under applicable terms of use (ToU)? If so, please describe this license and/or ToU, and provide a link or other access point to.**

Our benchmark dataset is released under a CC BY-NC 4.0 International License: `https://creativecommons.org/licenses/by-nc/4.0`. Our code implementation is released under the MIT License: `https://opensource.org/licenses/MIT`.

**Have any third parties imposed IP-based or other restrictions on the data associated with the instances? If so, please describe these restrictions, and provide a link or other access point to, or otherwise reproduce, any relevant licensing terms, as well as any fees associated with these restrictions.**

Yes, for commercial use, please check the website: `https://www.ncei.noaa.gov/`.

**Do any export controls or other regulatory restrictions apply to the dataset or to individual instances? If so, please describe these restrictions, and provide a link or other access point to, or otherwise reproduce, any supporting documentation.** No.

### B.7 MAINTENANCE

**Who will be supporting/hosting/maintaining the dataset?**

The authors of the paper.

**Is there an erratum? If so, please provide a link or other access point.**

Users can use GitHub to report issues or bugs.

**Will the dataset be updated (e.g., to correct labeling errors, add new instances, delete instances)? If so, please describe how often, by whom, and how updates will be communicated to dataset consumers (e.g., mailing list, GitHub)?**

Yes, the authors will actively update the code and data on GitHub. Any updates of the dataset will be announced in our GitHub repository.

**If the dataset relates to people, are there applicable limits on the retention of the data associated with the instances (e.g., were the individuals in question told that their data would be retained for a fixed period of time and then deleted)? If so, please describe these limits and explain how they will be enforced.**

The dataset does not relate to people.

**Will older versions of the dataset continue to be supported/hosted/maintained? If so, please describe how. If not, please describe how its obsolescence will be communicated to dataset consumers.**

Yes, we will provide the information on GitHub.

**If others want to extend/augment/build on/contribute to the dataset, is there a mechanism for them to do so? If so, please provide a description. Will these contributions be validated/verified? If so, please describe how. If not, why not? Is there a process for communicating/distributing these contributions to dataset consumers? If so, please provide a description.**

Yes, we welcome users to submit pull requests on GitHub, and we will actively validate the requests.

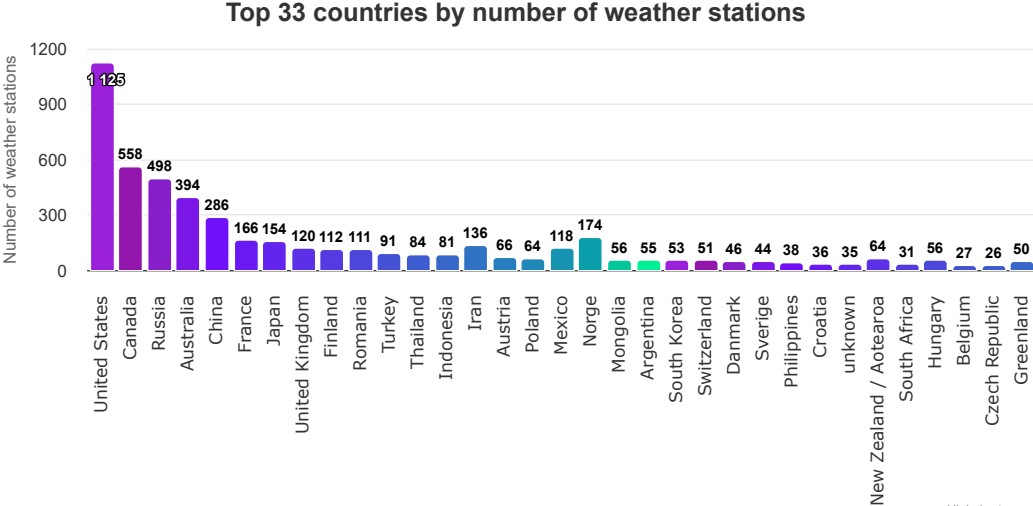

Figure 5: Statistics on the Number of Weather Stations in Different Countries and Regions.

## C    MORE DATASET ANALYSIS

**Station distribution by countries**. Figure 5 shows the histogram of the number of weather stations in the WEATHER-5K dataset over 33 countries. WEATHER-5K is a global database, though the best spatial coverage is evident in North America, Europe, Australia, and parts of Asia. Coverage in the Northern Hemisphere is better than the Southern Hemisphere.

**Compare ISD with MADIS Data Source**. Overall, as shown in Table 4, ISD is more diverse and offers broader coverage. Specifically, the surface data in MADIS mainly includes METER [2] and Mesonet [3]. Its reports primarily come from the U.S. while ISD collects surface weather data from more than 35,000 stations worldwide. Additionally, ISD spans a longer period from '1901 to the present' and is fully public for users. In future research, we believe including MADIS for station-based weather forecasting will further enhance this field.

Table 4: Differences between ISD (database of WEATHER-5K) and MADIS

| Dataset | Availability | Data Source | Time | Coverage |
|---------|-------------|-------------|------|----------|
| MADIS (METER and Mesonet) | Restricted | ASOS, AWOS, Airport Reports, CWOP, FAWH, GPSMet, KSDOT, RAWS, UDFCD, GLDNWS, IADOT, INTERNET | 2001-present | Primarily in U.S. |
| ISD | Fully public | More than 100 original data sources | 1901-present | 35,000 global stations |

Table 5: Statistics on weather station data

|  | Temperature | Dewpoint | Wind Direction | Wind Speed | Sea Level Pressure |
|--|-------------|----------|----------------|------------|---------------------|
| Mean | 12.71 | 6.53 | 191.19 | 3.37 | 1014.85 |
| Standard Deviation | 13.08 | 12.14 | 99.67 | 2.66 | 9.17 |

**Characteristics of data distribution**. Figure 6 provides violin plots for several variables. For temperature and dewpoint, the distributions of their data have similar shapes. The upper and lower distributions of the data are symmetrical around the median. The temperature distribution is most

---

[2]https://madis.ncep.noaa.gov/madis_metar.shtml
[3]https://madis.ncep.noaa.gov/madis_mesonet.shtml

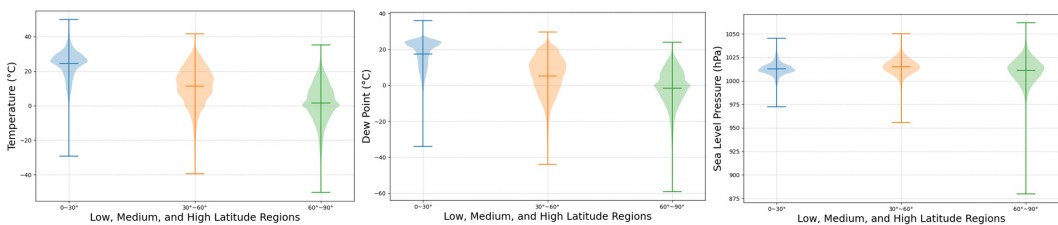

Figure 6: The violin plots of the temperature, dewpoint, and sea level pressure observations in the low, middle, and high latitudes, respectively. categorized by these two features.

concentrated in the low-latitude regions. As the latitude increases, the center of the temperature distribution starts to shift and also becomes more dispersed. This indicates that the temperature difference is larger in the mid-to-high latitude regions. For sea level pressure, we find that the distribution centers are similar across different latitudes, with little shift. However, as the latitude increases, the dispersion of sea level pressure becomes greater.

**Climate mean and standard deviation**. Table 5 presents the mean and standard deviation values for five key weather variables measured at 5,762 weather stations. The variables included are temperature, dewpoint, wind direction, wind speed, and sea level pressure. The mean temperature across the weather stations is 12.71 degrees, with a standard deviation of 13.08 degrees. For dewpoint, the mean is 6.53 degrees and the standard deviation is 12.14 degrees. The mean wind direction is 191.19 degrees, with a standard deviation of 99.67 degrees. The mean wind speed is 3.37 meters per second, with a standard deviation of 2.66 meters per second. Finally, the mean sea level pressure is 1014.85 millibars, with a standard deviation of 9.17 millibars. These statistics provide a high-level overview of the typical weather conditions captured by the network of weather stations.

Table 6: Table 3: Efficiency comparisons. Statistics are tested based on the following setting: batch size →1024 in general except 5600 for Corrformer, train time→estimated training time (in hours) for 300,000 iterations. task setting→48 input length and 120 output length. Hardware→224 Intel(R) Xeon(R) Platinum 8480CL CPUs @ 3.80 GHz, 2.0 TB RAM computing server, equipped with 8 NVIDIA H800 GPUs. Each experiment is conducted on a single GPU. Note that the total training time is also influenced by data read speed as some methods may suffer input starvation.

|  | Informer | Autoformer | Pyraformer | FEDformer | DLinear |
|---|---|---|---|---|---|
| Training Time (Hours) | 21∼22 | 36∼40 | 20∼21 | 38∼40 | 1.0∼1.5 |
| GPU Memory (MiB) | 12,880 | 64,688 | 33,750 | 18,804 | 850 |
| Parameters (M) | 11.32 | 10.53 | 7.54 | 16.29 | 0.01 |

|  | PatchTST | Corrformer | Mamba | iTransformer |  |
|---|---|---|---|---|---|
| Training Time (Hours) | 7∼8 | 144∼168 | 3∼4 | 3∼4 |  |
| GPU Memory (MiB) | 22,512 | 46,486 | 11,406 | 45,672 |  |
| Parameters (M) | 6.65 | 666.12 | 0.12M | 4.8 |  |

# D  MORE EXPERIMENTAL RESULTS

**Efficiency comparisons**. We summarize the efficiency comparisons in Table 6 with the following observations. In terms of training time, DLinear stands out as the most efficient, requiring only 1.0-1.5 hours for 300,000 iterations, while Mamba and iTransformer also demonstrate relatively fast training times of 3 and 4 days, respectively. On the other hand, Corrformer has the longest training time, taking 144-168 days. The other methods, Informer, Autoformer, Pyraformer, and FEDformer, have training times in the range of 20-40 hours. Regarding GPU memory usage, DLinear has the lowest requirement at 850 MiB, while Informer, Pyraformer, and Mamba have moderate GPU memory needs. Autoformer, FEDformer, Corrformer, and iTransformer, on the other hand, have relatively high GPU memory requirements, ranging from 18,804 MiB to 64,688 MiB. The trade-off between training time and GPU memory usage should be considered when selecting the

appropriate time-series forecasting method for a specific application, depending on the available computing resources and the requirements of the task.

**Visualization results**. In Figures 8 9 10 11 12 13 14, we have plot visualization results to showcase the performance of various time-series forecasting methods, including Pyraformer, FEDformer, DLinear, PatchTST, Mamba, iTransformer, and Corrformer. These visualizations provide a comparative analysis of how each of these different forecasting approaches performs on the time-series data.

By presenting the results in this series of figures, we are able to illustrate the unique characteristics and capabilities of each method. This allows the reader to gain a better understanding of the strengths and weaknesses of the various techniques, and how they may be suited for different types of time-series forecasting problems.

## E    LIVE WEATHER DEMON

In addition to providing high-quality datasets and benchmarks to promote scientific research, we are also committed to putting the research results into practice and providing weather services to the public. Figure 7 is a demo we are currently testing internally for providing weather station forecasts, and we will publish it in a GitHub repository in the future. The demo shows weather forecasts from the forecasting model trained on the WEATHER-5K dataset. The first data row shows the latest observation, the rest the forecast for the upcoming 24 hours.

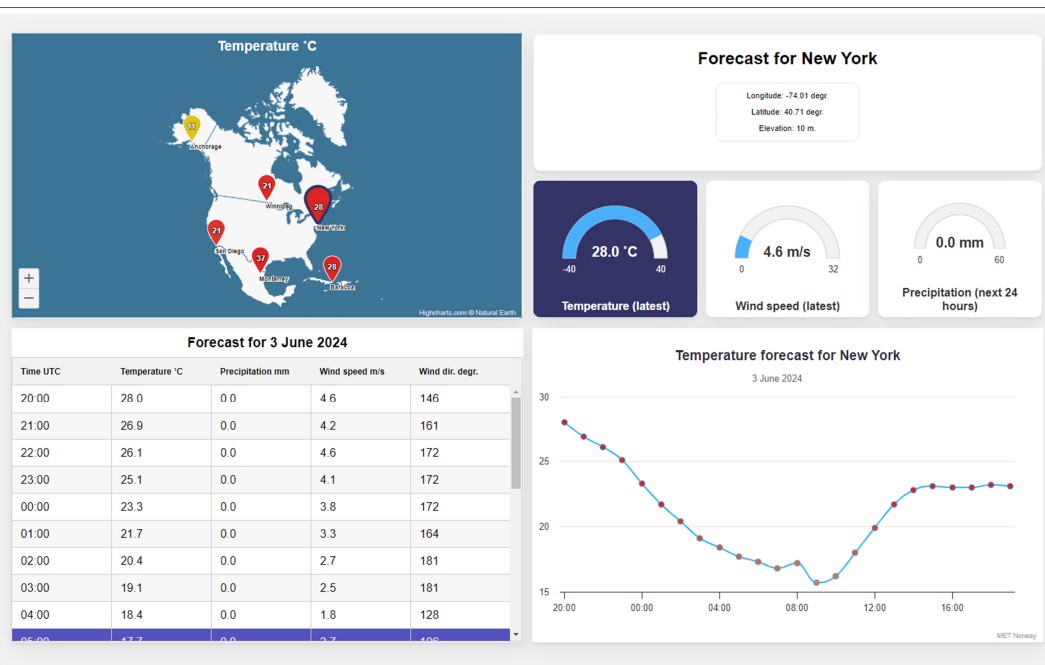

Figure 7: A live weather demo for global station weather forecasting.

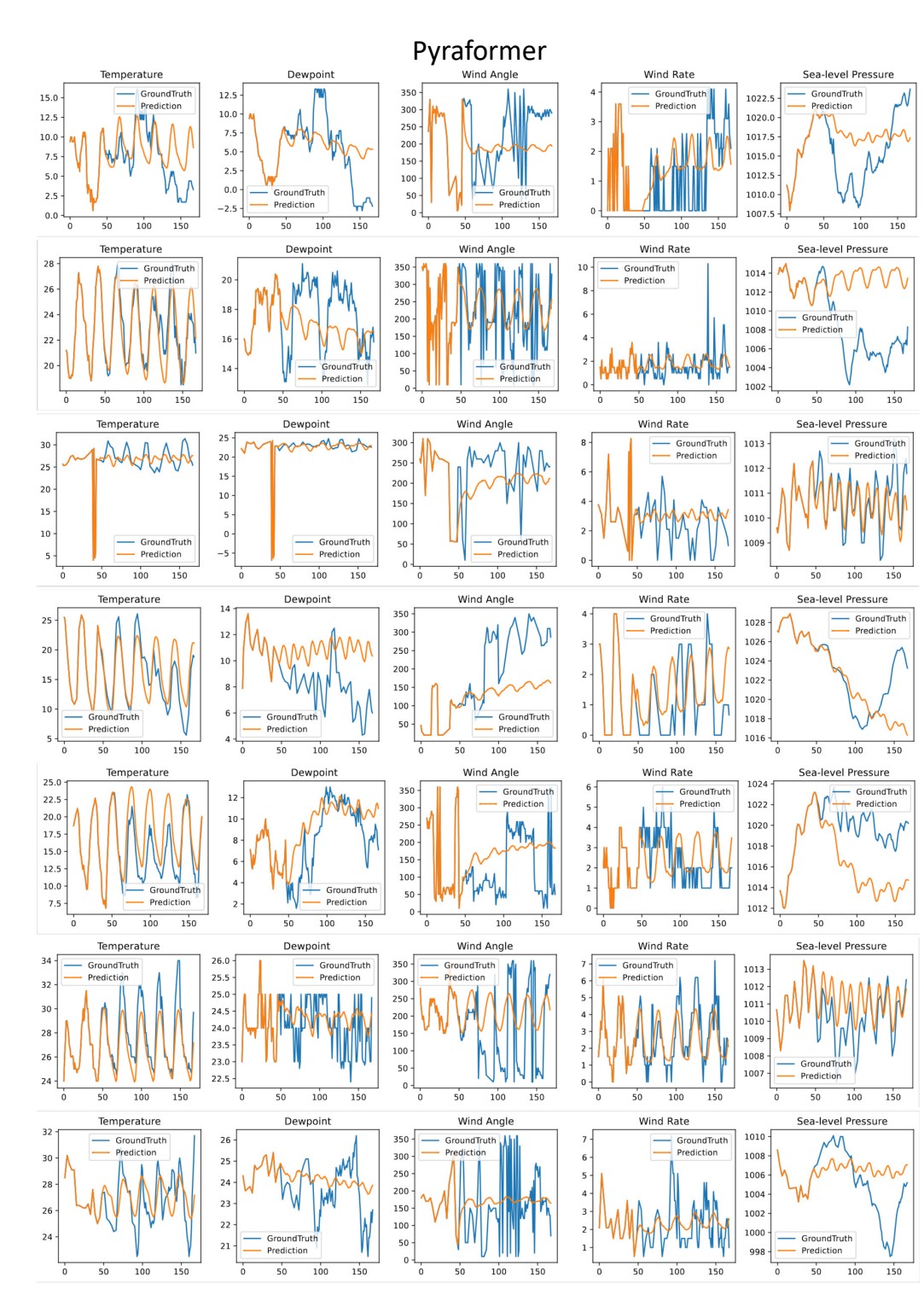

Figure 8: Visualization results of Pyraformer. Samples are randomly chosen. Orange lines are ground truths and Blue lines are predictions.

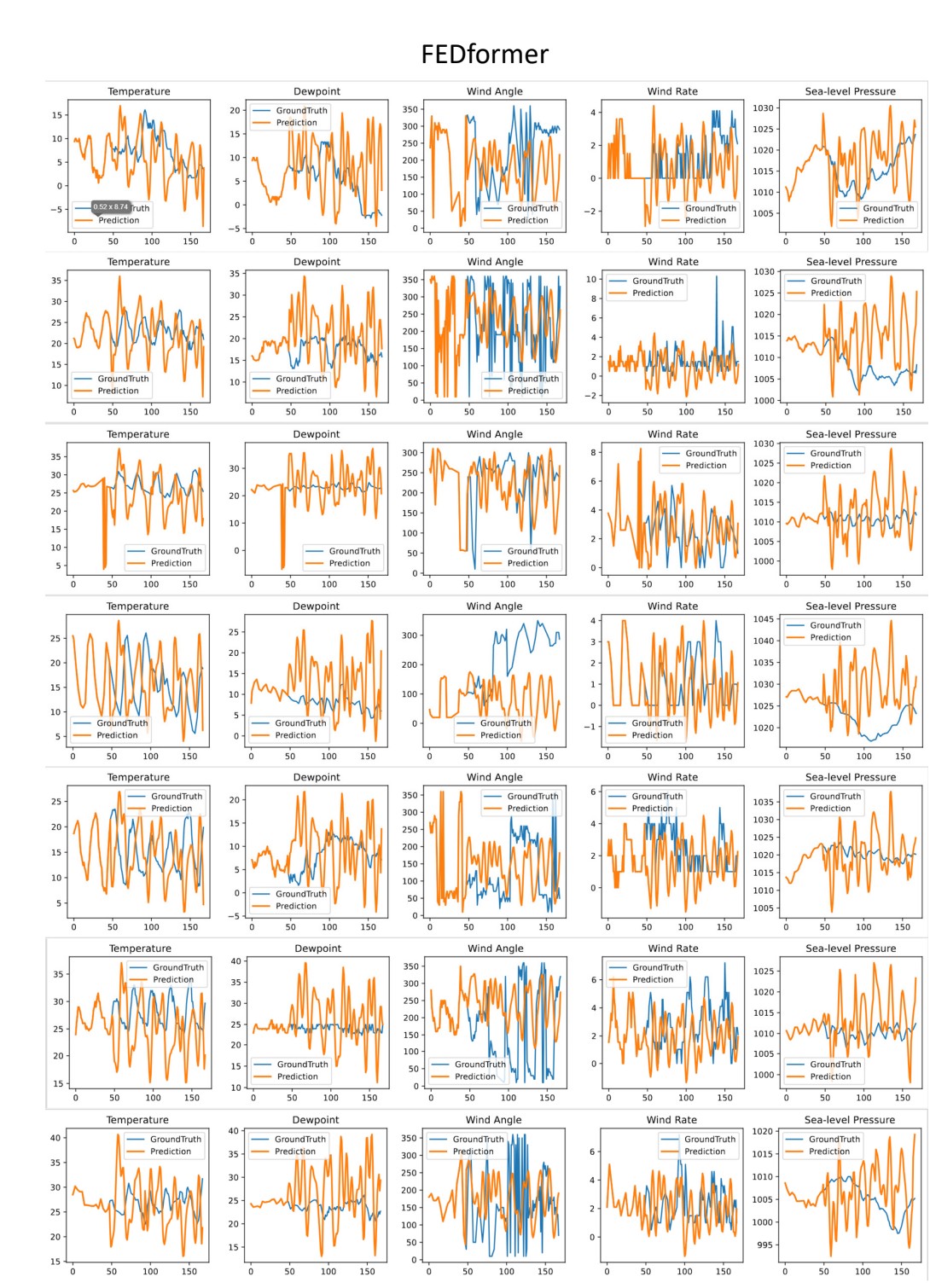

Figure 9: Visualization results of FEDformer. Samples are randomly chosen. Orange lines are ground truths and Blue lines are predictions.

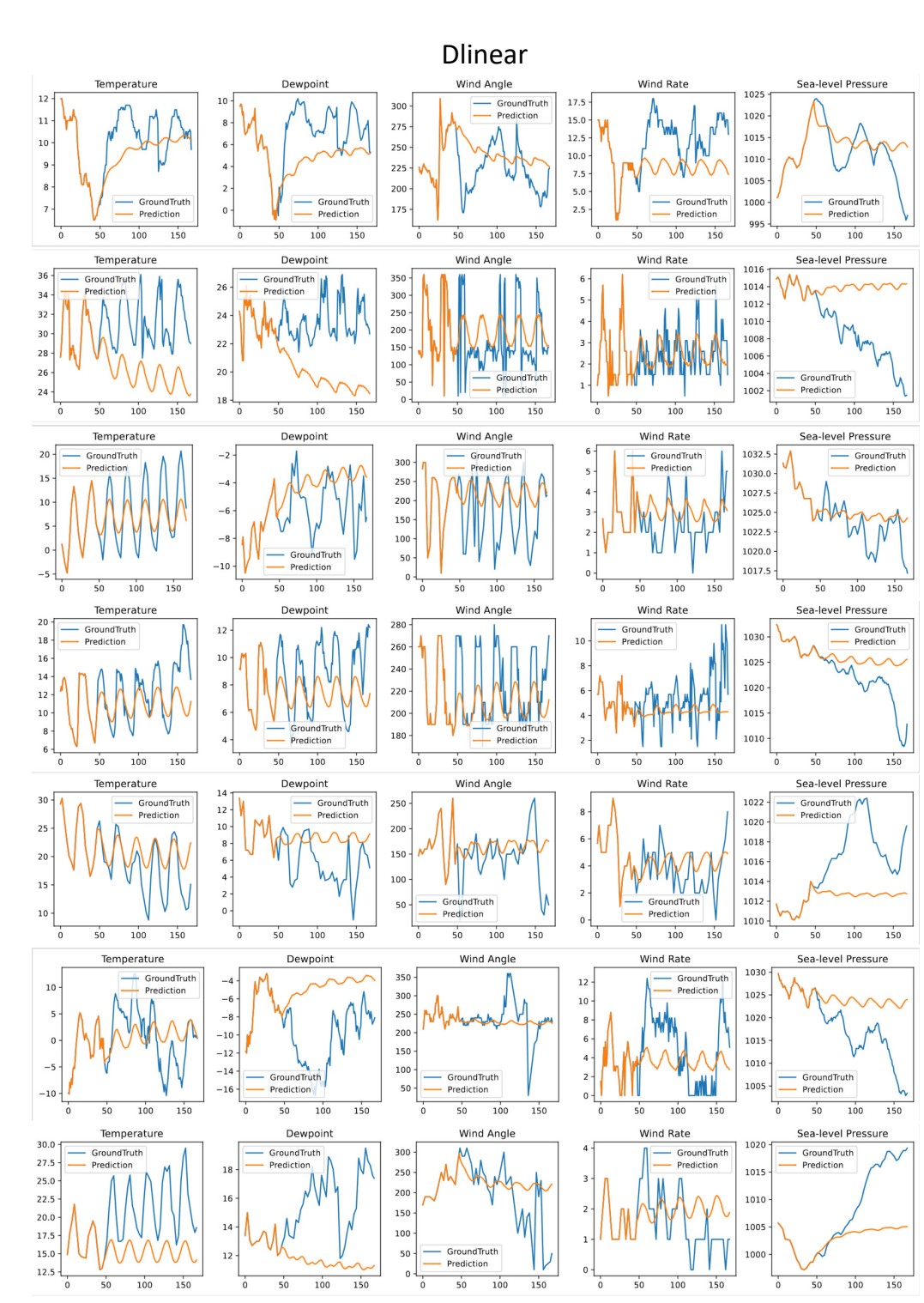

Figure 10: Visualization results of Dlinear. Samples are randomly chosen. Orange lines are ground truths and Blue lines are predictions.

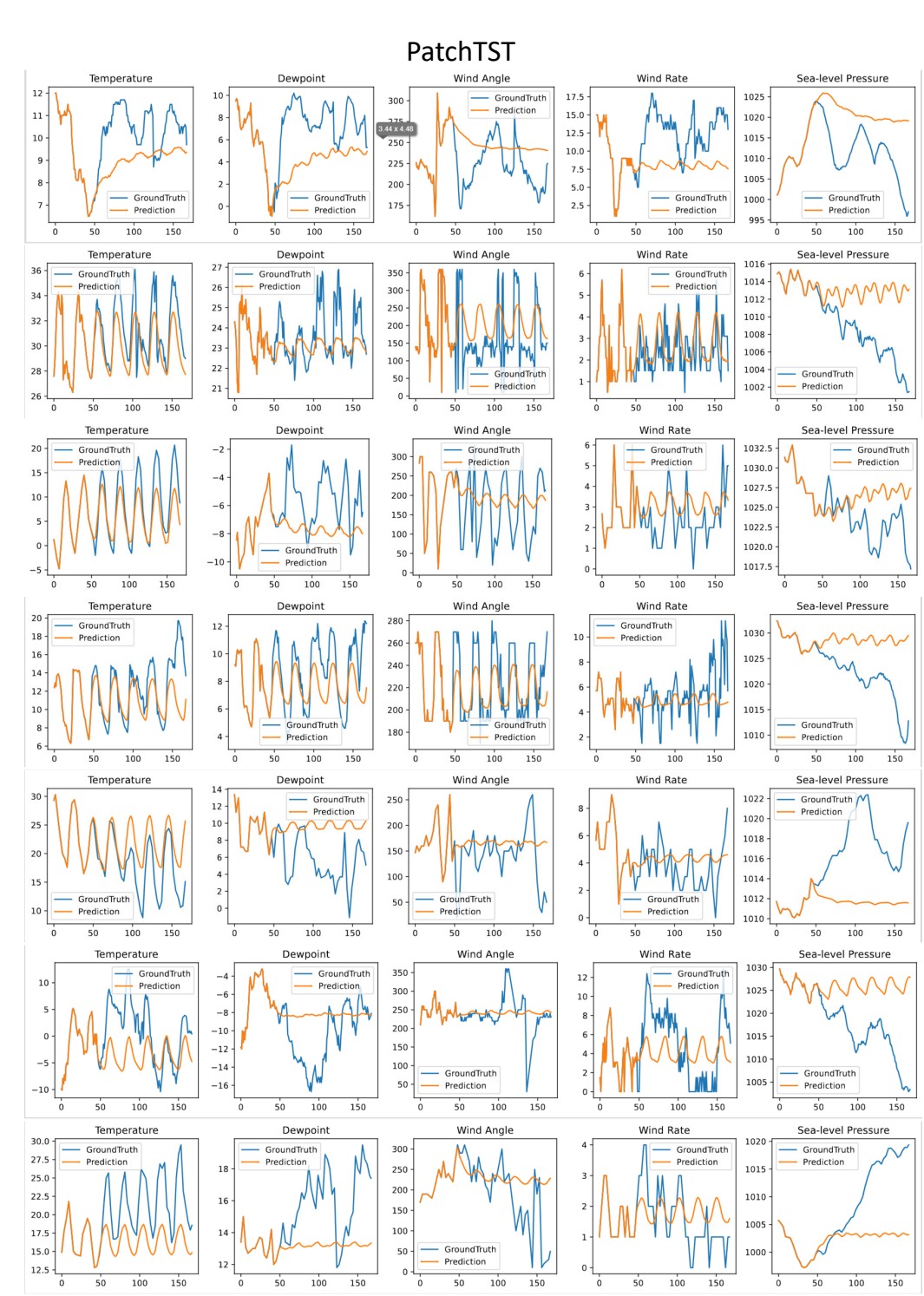

Figure 11: Visualization results of PatchTST. Samples are randomly chosen. Orange lines are ground truths and Blue lines are predictions.

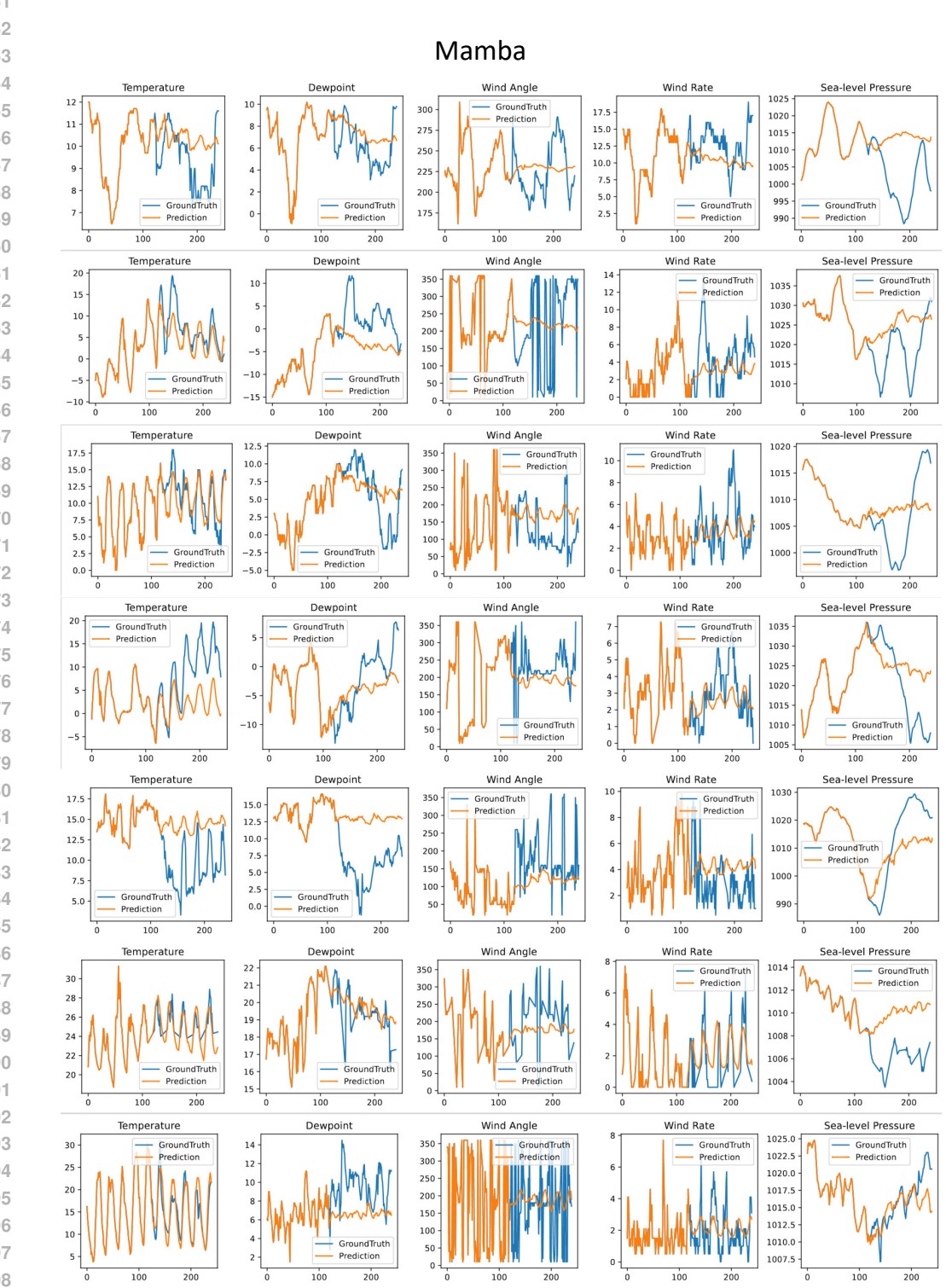

Figure 12: Visualization results of Mamba. Samples are randomly chosen. Orange lines are ground truths and Blue lines are predictions.

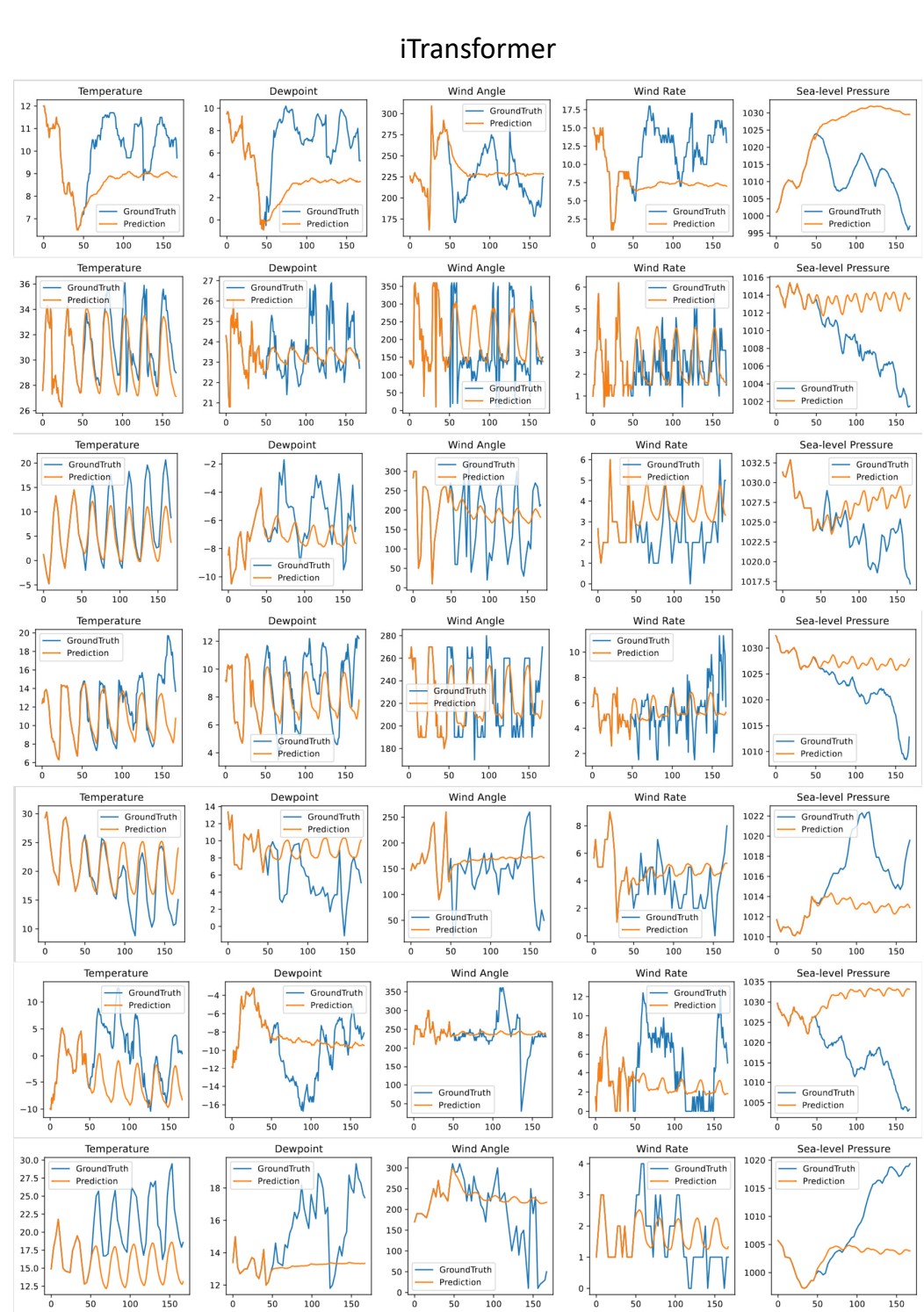

Figure 13: Visualization results of iTransformer. Samples are randomly chosen. Orange lines are ground truths and Blue lines are predictions.

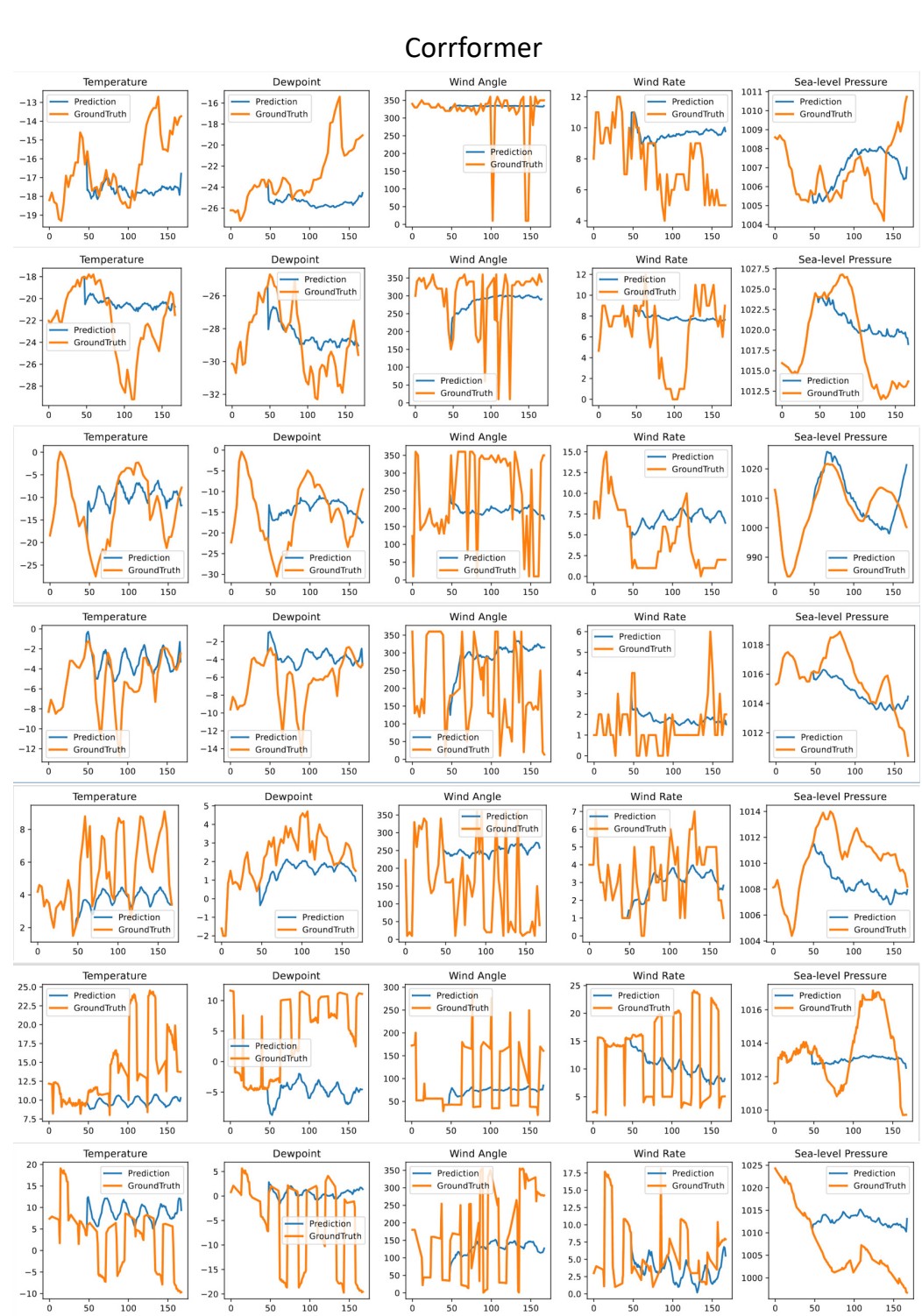

Figure 14: Visualization results of Corrformer. Samples are randomly chosen. Orange lines are ground truths and Blue lines are predictions.

