# OpenReview forum: "How Far are Today's Time-Series Models from Real-world Weather Forecasting Applications?"
_ICLR.cc/2025/Conference — Submitted to ICLR 2025_

### Official Review · Reviewer_SMXu · 2024-11-01

**Soundness:** 2
**Presentation:** 1
**Contribution:** 3
**Rating:** 3
**Confidence:** 5

**Summary:**

In this paper, the authors propose a novel weather dataset that can be utilized in various time series forecasting tasks. The dataset comprises data from over 5,000 weather stations worldwide, each providing five weather indicators, making it a promising resource. The authors have meticulously selected stations with 10 years of data and have cleaned the raw data from ISD using interpolation and ERA5 to ensure a complete dataset for future users.

The dataset is used to benchmark state-of-the-art transformer and state-space models against a Numerical Weather Prediction technique.

**Strengths:**

- Intensive work in dataset creation.
- Proposal of both a dataset and a benchmark.
- Data splitting follows the longest cycle of the data (one year).

**Weaknesses:**

- Lack of coherence and clarity in the presentation.
- Omission of some important, well-known datasets.
- Overgeneralization based on unique observations.
- Limited reproducibility.
- A thorough proof-reading is required.

**Questions:**

## Questions

### Q1
> $L=48$ for $H \in$ {$24,72,120,168$}

The experiment setting is not optimal for long-term time series forecasting (TSF), where transformer models typically excel. For shorter horizons, graph neural networks might be more suitable. The current setting, with input lengths of $48$, may not fairly represent the performance of transformer models, which are designed for larger input lengths (usually $96$ or $336$ in the literature). Including results for different input lengths, especially $96$, would enhance the benchmark.

The current state-of-the-art for long-term TSF is Pathformer. It would be interesting to see its performance on short-term forecasts.

### Q2
ISD data from NCEI/NOAA is the same source as the Corrformer and Informer weather datasets. Why are these datasets not included in Table 1? Although Weather from Informer is only one station and Global from Corrformer might have limited weather indicators, it is important to mention them to provide a complete picture and justify why WEATHER-5K should be preferred (cf. clarity section).

### Q3
What is the definition of an outlier, and how do the authors differentiate outliers from extreme weather? This distinction is crucial for users to understand modifications and potential model behavior. Have the modified timesteps been flagged for better model interpretation?

### Q4
> […] which highlights an unacceptably high RMSE when comparing ERA5 data to real-world observations […]

Are the authors suggesting that ERA5 is inaccurate? To the best of my knowledge, ISD measurements involve both manual and automated processes, which could introduce human error, especially in manual records, which will not be in favor to ISD based dataset (even with errors correction, which might introduce different types of errors).

### Q5
> Figur 3 e) illustrates the daily temperature variations at Chongqing city in China (57516099999) from June to September in 2022.

How many stations are included in this city? And so, is this relevant to use it as the main example?

### Q6
> This indicates that ERA5 consistently underestimated the diurnal temperature range at this station throughout the heatwave period.

This observation is based on one station over six months, compared to more than 5 000 stations over 10 years of data. Such a claim requires more extensive analysis. The authors seem to discredit ERA5 while earlier referring it as one of the most accurate real-time forecasting products and using it to fill missing data. Why use ERA5 if it is later discredited? If ERA5 is the most accurate, why not use it directly, especially given the potential for errors in ISD's manual records, extensive work for correction and data coverage limitation?

How many timesteps are considered extreme weather, and what percentage do they represent over the 10-year span? Is this percentage significant enough to promote WEATHER-5K over ERA5?

### Q7
In Figure 4a, where is the NWP model? The authors mention the computational cost of NWP models as a limitation but do not provide performance data. The parameters size /cost of some models, like PatchTST, known for high computational cost, seems inconsistent with the literature, which might advocate that the experiment setting (input of $48$) is insufficient to demonstrate predictive power. There is also an issue with the figure: Dlinear should be a point due to the parameter's size of $0.01$ but is not, and there are two circles without text (red and purple). What do they represent? Mamba and Autoformer? What do the training cost and error dotted lines represent?

### Q8 - Experiment setting
Early stopping of 3 is, in my experience, too low; 5 or 10 would be more appropriate.

### Q9 - Reproducibility
What is the batch size of Corrformer?

## Dataset

### D1
What is the point of repeating latitude and longitude for each row of a given weather station? This increases the dataset size unnecessarily unless stations are moving.

### D2
> we have used latitude, longitude, and elevation to represent their geographic locations.

Where is the elevation information? The instance field does not mention elevation in the CSV files.

### D3
> However, the proportion of error introduced by the interpolation is relatively small.

Authors need to provide the number of errors corrected and a mask or label column to identify these corrected timesteps. This information is crucial for many tasks and could open the dataset to other applications beyond forecasting.

## Limitations

### L1
Are errors always isolated timesteps on only one variable? If so, interpolation is understandable. If not, especially with consecutive timesteps with errors or missing data on multiple variables, interpolation is limited. In such cases, did the authors use ERA5? If so, make it clearer in the paper, and explain more in details how interpolation from ERA5 are made.

### L2
Not all users of this dataset will require worldwide stations for their applications. Have authors provided a way to select a subset of the dataset?
If not, please provide explanation of the weather stations name/ID formatting to simplify the selection task.

## Clarity / Coherence

### C1
Clarify the nature of Weather-5K and the targeted task. In my understanding, it is an hourly multivariate spatio-temporal weather dataset (several stations, each providing time series of several weather indicators) for short-, mid-, and long-term Time Series Forecasting. Forecasting tasks can be done in various fashions, such as:
- Univariate-to-Univariate (U), ex. Predict temperature of station XX for the next day using historical temperature of station XX
- Multivariate-to-Univariate (MU), ex. Predict temperature of station XX for the next day using historical weather indicators of station XX
- Multivariate-to-Multivariate (M), ex. Predict temperature of station XX for the next day using historical weather indicators of station XX
- Spatio-Temporal (ST) U
- ST MU, ex. Predict temperature of station XX for the next day using historical temperature from station AA to station WW
- ST M
- Multi-Variables (V) ST MU
- V ST M

Table 1 does not fully capture the dataset's potential in terms of forecasting tasks.
In addition, it should be revised:
- Use commas consistently for thousands.
- If "frequency" and "year" are provided, is "length" necessary? A column for missing data would be more informative.
- For the Exchange dataset, should $8$ be the value for "station" instead of "variable" since each column represents a country?

### C2
> Task settings

What is the forecasting fashion? M or MU? This is important for reproducibility. If M, are all stations considered, hence the input length of 48 "to balance computation and performance"? it not all stations, which subset of stations is used (for reproducibility)? If MU, which station is the target? And which stations are the inputs (all or a subset)?

### C3
> While they provide a solid foundation, their performance may be limited when faced with complex patterns or nonlinear relationships.

Do authors have a reference for this assumption?

### C4
> ML methods […] offer enhanced capabilities to handle nonlinear relationships and complex patterns.

Need references, perhaps [1] and [2].

[1] https://people.math.sc.edu/devore/publications/NLACTA.pdf
[2] https://link.springer.com/book/10.1007/978-3-319-58795-0

### C5
In section RW, paragraph "Data-driven numerical weather prediction," reintroduce the NWP acronym, the same should be done for GSWF to improve clarity.

### C6
For Figures 8 to 14, provide weather station IDs and sample IDs or origin forecast (t) for each row (for reproducibility). In addition, to highlight the necessity of WEATHER-5K, use samples depicting different scenarios, notably including extreme weather, to visually demonstrate model behavior. Therefore, identify and provide cases where extreme weather:
- Is in the input but not the predicted window.
- Is in the predicted window but not the input.
- Is in both.

Furthermore, in these figures, there is no need to repeat the title of each plot and the x-axis; use a 7 x 5 grid with the sharex option.

## Proof-read
To cite but a few:

1. **[Very important]** There is a formatting issue with citations in the first paragraph of the introduction and throughout the paper. Citations should be in parentheses unless the authors' names are part of the text, as per ICLR guidelines:
> When the authors or the publication are included in the sentence, the citation should not be in parenthesis using \citet{} (as in “See Hinton et al. (2006) for more information.”). Otherwise, the citation should be in parenthesis using \citep{} (as in “Deep learning shows promise to make progress towards AI (Bengio & LeCun, 2007).”)

2. Thie following text is repeated twice in the paper; remove the unnecessary repetition.
> 5, 672 weather stations worldwide are selected, spanning the period from 2014 to 2023, ensuring a recent and relevant time frame. This selection process focused on balancing the longevity of station operation, hourly data availabilit
y, and the inclusion of diverse weather variables.


Typos:
- “[…] into physically-based NWP models” or “physical-based” should be physics-based, no?
- “(with yea 2022)”
- “[…] benchmark experiments on WEATHER-5K,” comma?
- “[…] perform better. In our benchmarks. So developing efficient time-series […]”
- “[…] are may not be the optimal solution […]”
- “[…] except for Correformer.”
- “Figur 3 e) illustrates the daily temperature […]”
- “[…] of the WEATHER-5 dataset […]”

---

### Official Review · Reviewer_3akF · 2024-11-03

**Soundness:** 3
**Presentation:** 3
**Contribution:** 3
**Rating:** 6
**Confidence:** 3

**Summary:**

This paper introduces the WEATHER-5K dataset, a large-scale global station weather dataset designed to address the limitations of existing time-series forecasting (TSF) datasets.
The authors present WEATHER-5K, which includes comprehensive observational data from 5,672 global weather stations with hourly data spanning ten years.
The paper conducts extensive benchmarking of various TSF models against operational Numerical Weather Prediction (NWP) models. This comparison highlights the performance gap between academic TSF models and real-world weather forecasting.
Furthermore, a standardized evaluation framework and new metric (SEDI) are proposed for assessing TSF models, focusing on overall accuracy and extreme weather event prediction.

**Strengths:**

S1.
The paper presents an advancement in the field of time-series forecasting by introducing the WEATHER-5K dataset, which fills a notable gap in the availability of comprehensive, high-quality weather data for model training and evaluation.

S2.
The authors demonstrate a thorough understanding of existing datasets' limitations and the challenges TSF models face in practical applications.
The methodology for constructing the WEATHER-5K dataset includes rigorous data selection, quality control, and pre-processing.

S3.
The paper is well-structured and clearly articulated.

**Weaknesses:**

W1.
The benchmarking against NWP models primarily focuses on traditional NWP methods.
This paper could benefit from including a wider variety of contemporary data-driven NWP models, such as those utilizing machine learning techniques.

W2.
The performance analysis largely focuses on traditional metrics like Mean Absolute Error (MAE) and Mean Square Error (MSE), which, while standard, may not fully capture the complexities of weather forecasting, especially regarding extreme weather events.

W3.
This work does not adequately address the robustness of the proposed TSF models under varying conditions, such as different seasons or geographical variations.

**Questions:**

Q1.
How do you view the trade-offs between model complexity and interpretability in practical forecasting applications?

---

### Official Review · Reviewer_ioe6 · 2024-11-04

**Soundness:** 2
**Presentation:** 3
**Contribution:** 2
**Rating:** 5
**Confidence:** 3

**Summary:**

This paper proposes a new benchmark dataset, called Weather-5K, for evaluating general time series forecasting models on weather forecasting tasks. The dataset is built using data from a public database (ISD) and includes five types of hourly weather data from over 5,672 stations spanning 2014 to 2023, following quality control and post-processing. Experiments comparing general time series forecasting models with a Numerical Weather Prediction (NWP) model are conducted, with results analyzed to gain insights into the use of general time series forecasting models for weather forecasting tasks.

**Strengths:**

1. A new benchmark dataset is constructed specifically for station-based weather forecasting tasks.
2. The process of dataset construction is clearly described.
3. Experiments are conducted to compare general time series forecasting models with a specialized weather prediction model, providing insights into the limitations of general forecasting models for weather prediction and identifying future research opportunities.

**Weaknesses:**

1. Beyond computational complexity, are there other reasons why general time series forecasting models are needed for weather forecasting? Also, even regarding complexity, a comparison between general time series forecasting models and specific weather forecasting models should be provided to support this claim.
2. Some datasets, such as Weather-Australia, GlobalTempWind, and CMA_Wind, are not cited correctly.
3. In the outlier detection process, it is unclear how to ensure that detected outliers are not extreme weather events.
4. The paper states, “these models, operating at the mesh space (e.g., grid resolution of 0.25° and 0.09°), may not be the optimal solution for GSWF as discussed in Section 1”; however, there does not appear to be a related discussion in Section 1.
5. The statement "5,672 weather stations worldwide are selected..." is repeated on Page 4 and Page 5.
6. It is unclear whether RQ4 describes the bridge between TSF models and NWP models or between GSWF and NWP models.

**Questions:**

Same as the weaknesses.

---

### Official Review · Reviewer_P8nZ · 2024-11-09

**Soundness:** 2
**Presentation:** 2
**Contribution:** 2
**Rating:** 3
**Confidence:** 5

**Summary:**

The submitted paper is an interesting contribution in the field of time series forecasting for a specific problem domain (weather). The key innovation of the paper is 1) preparation of the dataset to be used for community 2) its benchmark over existing algorithms 3) and suggestion for the future work.

**Strengths:**

The dataset preparation (collection followed by data cleaning) is at the centre of contribution.

**Weaknesses:**

The paper miss the latest research in the domain of TSF, such as use of foundation model for long term forecasting. Such as
- Tiny Time Mixer
- MORAI
- Chronos
- ..
Please review literature and add methods.

Also, there are method for AutoAI, AutoARIMA and etc from statistical domain.

**Questions:**

None.

---

> ### Author Response · Authors · 2024-11-26
> **Your comments lack sufficient detail to support your conclusions**
>
> We are concerned about the quality of your review and the academic rigor demonstrated therein. It is surprising that your comments highlighted methods that were not implemented, while overlooking the key methods that were successfully realized in our work. Such a review approach, which appears to focus disproportionately on perceived shortcomings, raises questions about the depth of expertise required for a constructive and balanced review.
>
> We believe that maintaining the high standards of top-tier conferences requires reviewers to provide fair, insightful, and well-supported feedback, and we encourage you to approach future reviews with these principles in mind.

---

### Meta-Review · Area_Chair_8nK9 · 2024-12-21

**Metareview:**

This paper proposes a comprehensive time-series dataset, WEATHER-5K, for weather forecasting and compares general time-series forecasting (TSF) models with Numerical Weather Prediction (NWP) models to highlight the limitations of current TSF models.
The paper provides a valuable resource for researchers and introduces important new research challenges.
However, it falls short in adequately comparing with the latest studies and datasets, addressing limitations in evaluation metrics, ensuring reproducibility, and resolving inconsistencies in its arguments.
These areas need further improvement and are encouraged to be strengthened.

**Additional Comments On Reviewer Discussion:**

During the rebuttal phase, the authors addressed some points, including referencing recent research and clarifying explanations.
 (They also raised concerns about the quality of  a review.)
However, issues related to reproducibility and comparisons with the latest research remained unresolved, which means significant weaknesses have still been unaddressed.

---

### Decision · Program_Chairs · 2025-01-22

Reject